# Modeling corticotroph deficiency with pituitary organoids supports the functional role of *NFKB2* in human pituitary differentiation

Thi Thom Mac[1,2], Teddy Fauquier[1]*, Nicolas Jullien[3], Pauline Romanet[1,4], Heather Etchevers[1], Anne Barlier[1,4,5], Frederic Castinetti[1,5], Thierry Brue[1,5]*

[1]Aix-Marseille University, INSERM, UMR1251, Marseille Medical Genetics, Institut MarMaRa, Marseille, France; [2]Hanoi Medical University Hospital, Hanoi, Viet Nam; [3]Aix-Marseille University, CNRS, UMR7051, Institut de NeuroPhysiopathologie, Marseille, France; [4]Aix-Marseille University, APHM, INSERM, MMG, Laboratory of Molecular Biology, La Conception Hospital, Institut MarMaRa, Marseille, France; [5]Aix Marseille University, APHM, INSERM, MMG, Department of Endocrinology, La Conception Hospital, Institut MarMaRa, Marseille, France

*For correspondence:
teddy.fauquier@univ-amu.fr (TF);
Thierry.BRUE@ap-hm.fr (TB)

Competing interest: The authors declare that no competing interests exist.

## eLife Assessment

This **important** study examines the effects of NFKB2 mutations on pituitary gland development through hypothalamic-pituitary organoids. The evidence supporting the main conclusions is **solid**, although analysis of additional clones to exclude inter-clone variability would strengthen the conclusions. This is a revised study, but insight into the mechanism of action of NFKB2 during pituitary development is incomplete. This work will be of interest to endocrinologists and biologists working on pituitary gland development and disease.

**Abstract** Deficient Anterior pituitary with common Variable Immune Deficiency (DAVID) syndrome results from *NFKB2* heterozygous mutations, causing adrenocorticotropic hormone deficiency (ACTHD) and primary hypogammaglobulinemia. While NFKB signaling plays a crucial role in the immune system, its connection to endocrine symptoms is unclear. We established a human disease model to investigate the role of *NFKB2* in pituitary development by creating pituitary organoids from CRISPR/Cas9-edited human induced pluripotent stem cells (hiPSCs). Introducing homozygous *TBX19*^K146R/K146R^ missense pathogenic variant in hiPSC, an allele found in congenital isolated ACTHD, led to a strong reduction of corticotrophs number in pituitary organoids. Then, we characterized the development of organoids harboring *NFKB2*^D865G/D865G^ mutations found in DAVID patients. *NFKB2*^D865G/D865G^ mutation acted at different levels of development with mutant organoids displaying changes in the expression of genes involved on pituitary progenitor generation (*HESX1*, *PITX1*, *LHX3*), hypothalamic secreted factors (*BMP4, FGF8, FGF10*), epithelial-to-mesenchymal transition, lineage precursors development (*TBX19, POU1F1*) and corticotrophs terminal differentiation (*PCSK1, POMC*), and showed drastic reduction in the number of corticotrophs. Our results provide strong evidence for the direct role of *NFKB2* mutations in the endocrine phenotype observed in patients leading to a new classification of a *NFKB2* variant of previously unknown clinical significance as pathogenic in pituitary development.

## Introduction

ACTHD is defined by an insufficient production of ACTH by the pituitary, and then low adrenal cortisol production. Proper diagnosis and management of ACTHD is crucial as it is a life-threatening condition in the neonatal period characterized by hypoglycemia, cholestatic jaundice, and seizures (*Couture et al., 2012*). Constitutional ACTHD, i.e., ACTHD diagnosed at birth or during the first years of life, can be isolated or associated with other pituitary hormones deficiencies, such as growth hormone (GH) or thyrotropin-stimulating hormone (TSH), then as part of combined pituitary hormone deficiency (CPHD). ACTHD can be due to mutations in genes coding for transcription factors responsible for pituitary ontogenesis, especially the T-box transcription factor *TBX19* (also known as *TPIT*), *NFKB2, LHX3, LHX4, PROP1, HESX1, SOX2, SOX3, OTX2, and FGF8*.

Mutations of *TBX19* account for approximately two-thirds of neonatal-onset complete isolated ACTHD (*Couture et al., 2012*). TBX19 is a T-box transcription factor restricted to pituitary pro-opiomelanocortin (POMC)-expressing cells in mice and humans. It is essential for *POMC* gene transcription and terminal differentiation of POMC-expressing cells (*Lamolet et al., 2001*). POMC is the precursor of ACTH. *Tbx19*-deficient mice have only a few pituitary *Pomc*-expressing cells, with very low ACTH and undetectable corticosterone levels (*Pulichino et al., 2003*). In humans with isolated ACTH deficiency, *TBX19* mutations lead to loss-of-function by different mechanisms, such as the *K146R* (exon 2, c.437 A>G) pathogenic variant located in the T-box region, which results in a loss of DNA-binding ability (*Couture et al., 2012*).

Thanks to GENHYPOPIT (*Reynaud et al., 2006*; *Jullien et al., 2019*; *Jullien et al., 2021*), an international network aimed at identifying new genetic etiologies of combined pituitary hormone deficiency, we described the first cases of DAVID syndrome, a rare association of hypopituitarism (mainly ACTHD) and immune deficiency (hypogammaglobulinemia) (*Quentien et al., 2012*). DAVID syndrome is associated with variants in the nuclear factor kappa-B subunit 2 (*NFKB2*) gene (*Chen et al., 2013*; *Brue et al., 2014*). In particular, we reported that a pathogenic, heterozygous D865G (exon 23, c.2594 A>G) *NFKB2* variant was found in a patient presenting with severe recurrent infections from 2 y of age, who at the age of 5 was diagnosed with ACTHD (*Quentien et al., 2012*). Mutations in the C-terminal region of NFKB2 lead to the disruption of both non-canonical and canonical pathways (*Wirasinha et al., 2021*; *Kuehn et al., 2017*; *Lee et al., 2014*). Full-length NFKB2 (p100) protein has two critical serines, S866 and S870, in the C-terminal domain. Phosphorylation of these sites yields the transcriptionally active NFKB2 (p52) through proteasomal processing of p100 protein. The D865G mutation, located adjacent to the critical S866 phosphorylation site, protects mutant protein from proteasomal degradation, causes defective processing of p100 to p52, and results in reduced translocation of p52 to the nucleus (*Chen et al., 2013*; *Kuehn et al., 2017*; *Lee et al., 2014*). $NFKB2^{+/D865G}$ resulted in 50% of normal processing to p52, whereas $NFKB2^{D865G/D865G}$ exhibited near-absence of p52 (*Lee et al., 2014*). The nuclear factor kappa B (NFKB) signaling pathway is a known key regulator of the immune system (*Lee et al., 2014*; *Lindsley et al., 2014*; *Carragher et al., 2004*), which likely explains the immune phenotype of patients with DAVID syndrome, including susceptibility to infections and auto-immune disorders (*Mac et al., 2023*). In contrast, the underlying mechanism causing pituitary disorders remains unknown, with two predominating hypotheses: an indirect autoimmune hypophysitis, preferentially affecting corticotroph function, or a primary developmental defect suggested by the expression of *NFKB2* in the developing human pituitary (*Zhang et al., 2020*). However, a developmental role for *NFKB2* was not confirmed in the mouse: the *Lym1* mouse model carrying a homozygous nonsense variant Y868* presented an apparently normal pituitary development and function (*Brue et al., 2014*).

Over the last years, human induced pluripotent stem cell (hiPSC)-derived organoids have emerged as promising models to study many developmental mechanisms and their perturbation in disease (*Ho et al., 2018*). Pioneering work on human embryonic stem cells and later, hiPSC, has established that 3D organoid models can replicate aspects of pituitary development (*Ozone et al., 2016*; *Kasai et al., 2020*; *Matsumoto et al., 2019*), and could thus be of major interest in modeling pituitary disorders (*Ozone et al., 2016*; *Suga et al., 2011*; *Sasai et al., 2012*). In the present study, we applied recent progress in genome editing using CRISPR/Cas9 (*Sun and Ding, 2017*) to a refined protocol to derive 3D organoids from hiPSC in order to model ACTHD. We validated the recapitulation of corticotroph cell differentiation in vitro by introducing a *TBX19* mutation known to induce isolated human ACTHD and documenting the subsequent deficiencies in corticotroph development. When our model was

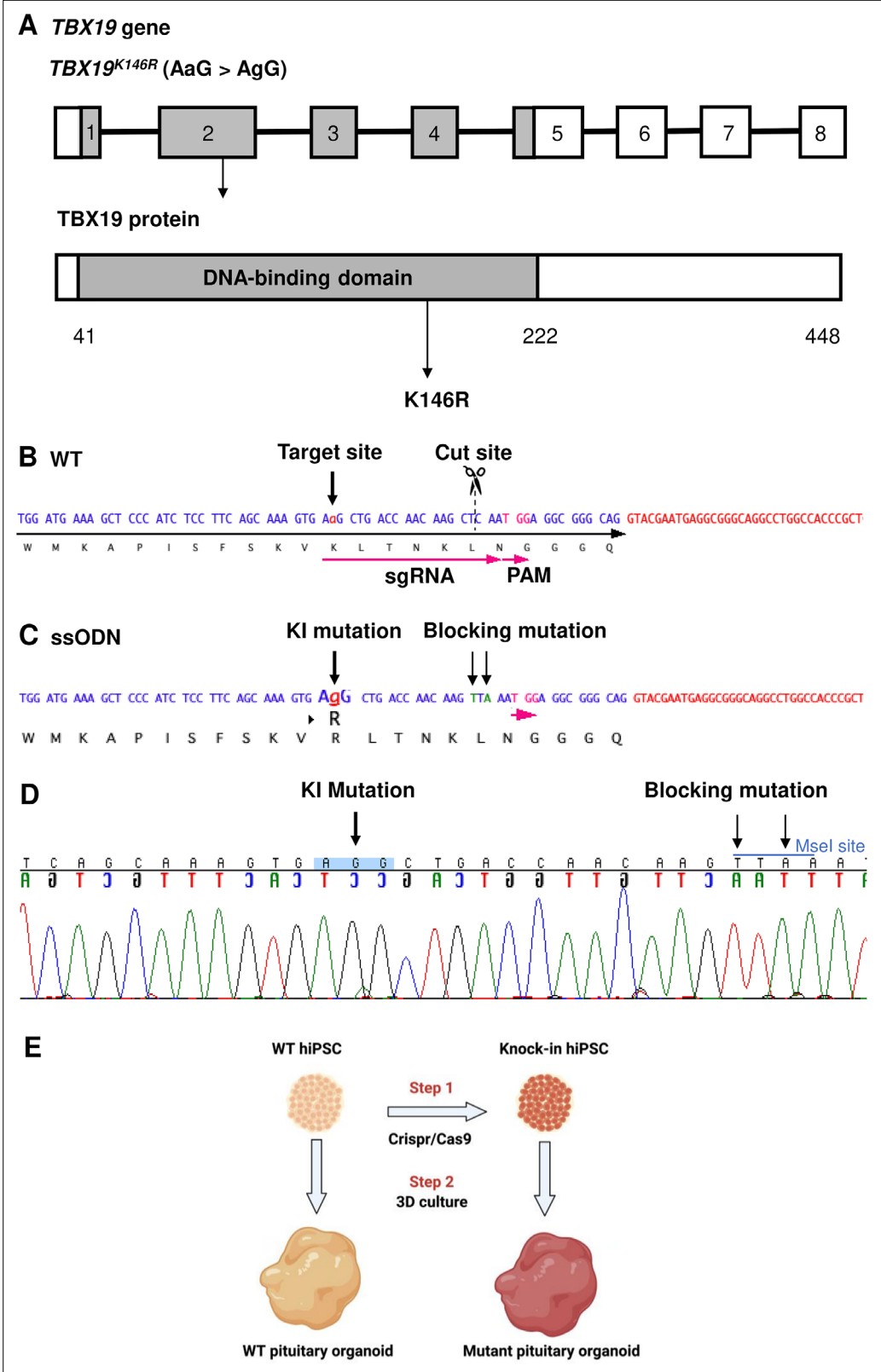

**Figure 1.** Design of single guide RNA (sgRNA) and single-stranded oligo DNA nucleotides (ssODN) to edit in the *TBX19^{K146R}* mutation. (**A**) Illustration of the *TBX19* gene (HGNC: 11596; ENSEMBL: ENSG00000143178; Human GRCh38) and TBX19 protein. (**B**) Wild-type (WT) sequence containing target site, cut site, target, and protospacer-adjacent motif (PAM) sequences. (**C**) ssODN design to edit in a missense K146R mutant of *TBX19* using CRISPR/

*Figure 1 continued on next page*

*Figure 1 continued*

Cas9. (**D**) Sequence analysis of a *TBX19* KI human induced pluripotent stem cell (hiPSC) clone 63 obtained by Sanger sequencing after screening by cleaved amplified polymorphic sequences (CAPS). This clone was subsequently used in this work to differentiate into pituitary organoids (see below). (**E**) Summary of our strategy procedure. Step 1: Production of the knock-in hiPSC lines by CRISPR/Cas9 genome editing. Step 2: Differentiation into pituitary organoids from mutant hiPSC lines in parallel with the isogenic WT line using 3D culture, followed by the comparison of the development of organoids between the two groups.

The online version of this article includes the following source data and figure supplement(s) for figure 1:

**Figure supplement 1.** Summary of the key steps of CRISPR/Cas9-mediated genome knock-in editing in human induced pluripotent stem cell (hiPSC).

**Figure supplement 2.** Results of cleaved amplified polymorphic sequences (CAPS) assay and Sanger sequencing analysis for editing *TBX19* mutation.

**Figure supplement 2—source data 1.** A pdf file describing modifications of original pictures used in *Figure 1— figure supplement 2*.

**Figure supplement 2—source data 2.** Three original pictures used in *Figure 1—figure supplement 2*.

used to characterize the D865G *NFKB2* variant found in patients with DAVID syndrome, it displayed dramatically altered corticotroph differentiation in the absence of immune cells, clearly demonstrating for the first time a direct role of *NFKB2* in human pituitary development.

## Results

## Corticotroph deficiency can be modeled by directed differentiation of *TBX19*-mutant hiPSC in 3D culture

To first validate the pituitary organoid model and determine whether the *TBX19* mutant can affect corticotrophs differentiation, we generated a *TBX19* KI hiPSC line from the control line using CRISPR/ Cas9 (*Figure 1A–D*; *Figure 1—figure supplement 1*). One KI clone carrying $TBX19^{K146R/K146R}$ was obtained after screening 100 clones by CAPS and confirmation by Sanger sequencing (*Figure 1— figure supplement 2*). This mutant clone was then amplified and differentiated into pituitary organoids, in parallel with the control line.

The ability of the *TBX19* KI line to differentiate into the hypothalamic-pituitary structure was compared to the control line (200 organoids for each line) using a 3D organoid culture method, in which pituitary-like and hypothalamus-like structures simultaneously develop (*Figures 1E and 2A*, *Figure 2—figure supplement 1A–B*). In this method, hiPSC differentiates into hypothalamic progenitors in the central part of the organoids, whereas the outer layer differentiates into oral ectoderm, that will in turn develop into anterior pituitary tissue (*Figure 2A*, *Figure 2—figure supplement 1C*). The expression of several key markers of pituitary development and differentiation was then compared in mutant *TBX19* KI organoids over time, matched to control WT organoids using qRT-PCR (d0, d6, d18, d27, d48, d75, d105) and immunofluorescence (d48 and d105) (*Figure 2A*). Organoids grew in the culture medium with average sizes in their greatest dimension from 0.4 mm on day 6–1.9 mm on day 105 (*Figure 2B*).

Successful differentiation was validated in WT organoids by qRT-PCR, at early stages by the expression of a set of pituitary transcription factors, including *HESX1*, *PITX1*, and *LHX3*, and at the latest stage of corticotroph cell differentiation by the expression of *TBX19* and *POMC* (*Figure 2A and C–H*). HESX1 is the first specific marker of pituitary primordium and its expression is important for the early determination of the gland. In WT organoids, *HESX1* peaked around day 6 of culture, and was rapidly downregulated from day 18 (*Figure 2D*). Expression of *PITX1* (oral ectoderm marker) and *LHX3* (pituitary progenitor marker) increased from day 27, then *PITX1* reached a plateau from d48 (*Figure 2E*) whereas *LHX3* peaked at d75 and was then slightly downregulated (*Figure 2F*). Immunofluorescence confirmed the presence of oral ectoderm cells expressing PITX1 and CDH1 (E-cadherin, *Figure 2— figure supplement 1C*), therefore, resembling Rathke's pouch progenitors. The effective generation of pituitary progenitors was confirmed by the presence of LHX3 + cells in WT organoids on day 48 (*Figure 3A*, *Figure 2—figure supplement 1A*). These cells were located in the outermost layer of WT organoids, surrounding the hypothalamic progenitor layer that expressed NKX2.1 (*Figure 2—figure*

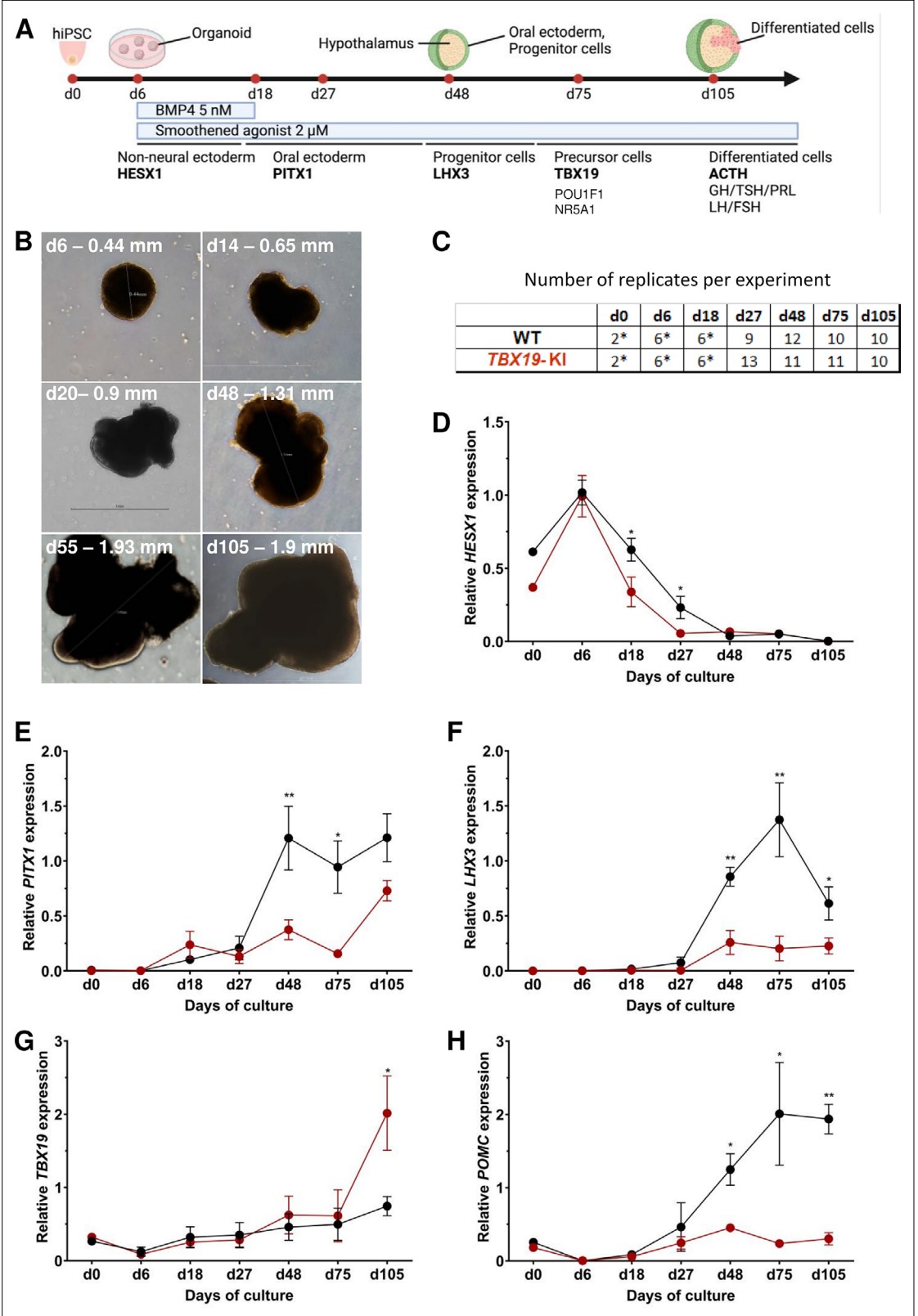

**Figure 2.** Time course of organoid growth and gene expression in wild-type (WT) and *TBX19* KI organoids. (**A**) Culture protocol and outline to generate pituitary organoids in three-dimensional (3D) culture from human induced pluripotent stem cell (hiPSC). Organoids were collected at days (**d**) 0, 6, 18, 27, 48, 75, and 105 during differentiation to analyze. (**B**) Brightfield microscopy views of WT organoid examples at different time points throughout differentiation. Scale bars are indicated in each image. (**C**) Number of replicates per time point and per genotype in experiments depicted in the

*Figure 2 continued on next page*

*Figure 2 continued*

following graphs. Asterisks indicate that 7–8 organoids were grouped for each sample. For other points, each sample consists of a single organoid. (**D–H**) Relative quantification (RQ) mRNA expression analysis for key markers of pituitary organoids during differentiation: WT (in black line) and *TBX19* KI organoids (in red line). Relative quantification of each target gene was obtained by the $2^{-\Delta\Delta Ct}$ method from qRT-PCR results (see Methods). Data show means ± standard error of the mean (SEM, Mann-Whitney t-test [unpaired, two-tailed, nonparametric]). p<0.05 (*), p<0.01 (**). (**D**) Relative quantification of *HESX1* expression, the earliest pituitary placode marker assessed. The expression of *HESX1* is significantly downregulated in *TBX19* KI organoids vs. WT at d18 and d27. (**E**) Relative quantification expression of *PITX1*, a pituitary progenitor marker. *PITX1* was significantly downregulated in *TBX19* KI organoids by d48 and d75. (**F**) Relative quantification expression of *LHX3*, a pituitary progenitor marker. *LHX3* was significantly lower in *TBX19* KI organoids as compared to WT from d48 onwards. (**G**) Relative quantification expression of *TBX19*, a critical transcriptional determinant for corticotroph differentiation. *TBX19* expression is higher in *TBX19* KI organoids at d105. (**H**) Relative quantification expression of pro-opiomelanocortin (*POMC*), a corticotroph marker. *POMC* was significantly downregulated in *TBX19* KI organoids from d48 onwards.

The online version of this article includes the following figure supplement(s) for figure 2:

**Figure supplement 1.** Differentiation of human induced pluripotent stem cell (hiPSC) control into pituitary organoid using 3D culture.

*supplement 1B*). By day 105, WT organoids contained many differentiated corticotrophs co-expressing TBX19 protein in the nucleus and ACTH in the cytoplasm, as seen in confocal microscopy, a critical feature for the rest of our investigations (n=8, *Figure 3B*, *Figure 2—figure supplement 1D*). Consistent with these images, qRT-PCR results confirmed the highest levels of *TBX19* and *POMC* expression after at least 75 d of culture in WT organoids (*Figure 2G–H*). Taken together, these data showed that pituitary organoids in 3D culture mimic human pituitary ontogenesis, and were characterized in our conditions by the ability to differentiate into pituitary progenitors by day 48 and into corticotroph cells by day 105.

We then tested whether ACTHD could be modeled by *TBX19*-mutant pituitary organoids. To this end, the development of pituitary organoids carrying *TBX19* KI was matched to WT organoid development and analyzed using qRT-PCR and immunofluorescence for pituitary ontogenesis markers as described above. Our data showed a significant decrease in the expression of *HESX1* at d18 and d27 in the mutant (*Figure 2D*). Both *PITX1* and *LHX3* transcript levels were significantly decreased by day 48 and day 75 in mutant organoids (*Figure 2E and F*), with only partial recovery for *PITX1* by d105, suggesting an impairment of pituitary progenitor generation. Although *TBX19* transcript levels were unchanged until d75 (*Figure 2G*), we observed a decrease in *POMC* expression in the *TBX19* KI organoids that was significant from day 48 onwards (*Figure 2H*). *TBX19* expression was significantly higher in mutant organoids at d105, but this had no influence on *POMC* expression, which remained very low. These results confirmed that our organoid model can recapitulate the need for a fully functional TBX19 protein to achieve proper *POMC* gene transcription during human pituitary development.

Next, we checked several pituitary markers by immunofluorescence (*Figure 3A–D*). In line with qRT-PCR results, *TBX19* KI organoid immunostaining on day 48 showed that LHX3 protein expression was decreased in *TBX19* KI organoids (n=10 organoids for each group, *Figure 3A and B*). By day 105, we observed that there were significantly fewer corticotroph cells expressing ACTH and TBX19 proteins in *TBX19*$^{K146R/K146R}$ organoids (n=10 organoids for each group, *Figure 3C and D*).

Finally, in order to take into account the possibility of regionalized sampling in the section, we performed a quantitative analysis of ACTH by transparizing organoids immunostained for the hormone and imaging them by light-sheet confocal microscopy. 3D reconstruction confirmed that ACTH was nearly absent from *TBX19* KI organoids on day 105 compared to control organoids (*Figure 4A–B* and *Figure 4—video 1* and *Figure 4—video 2*). We observed that corticotroph cell distribution was not uniform throughout the organoids, but was rather concentrated in one or a few buds. In summary, there was a significant decrease in the number of ACTH + corticotroph cells with a *TBX19* KI genotype on day 105 versus control organoids (*Figure 4C*).

Together, this first set of experiments demonstrated that genome editing of hiPSC with a KI homozygous pathogenic variant of *TBX19* effectively prevented their differentiation into ACTH-producing corticotrophs.

## NFKB2 signaling is vital for corticotroph development

As our organoid model for the study of ACTHD was validated with the *TBX19* mutant line, we then used the same approach to investigate the potential role of NFKB2 in human hypothalamic-pituitary development. To do so, we first established a *NFKB2* KI mutant hiPSC line using CRISPR/Cas9

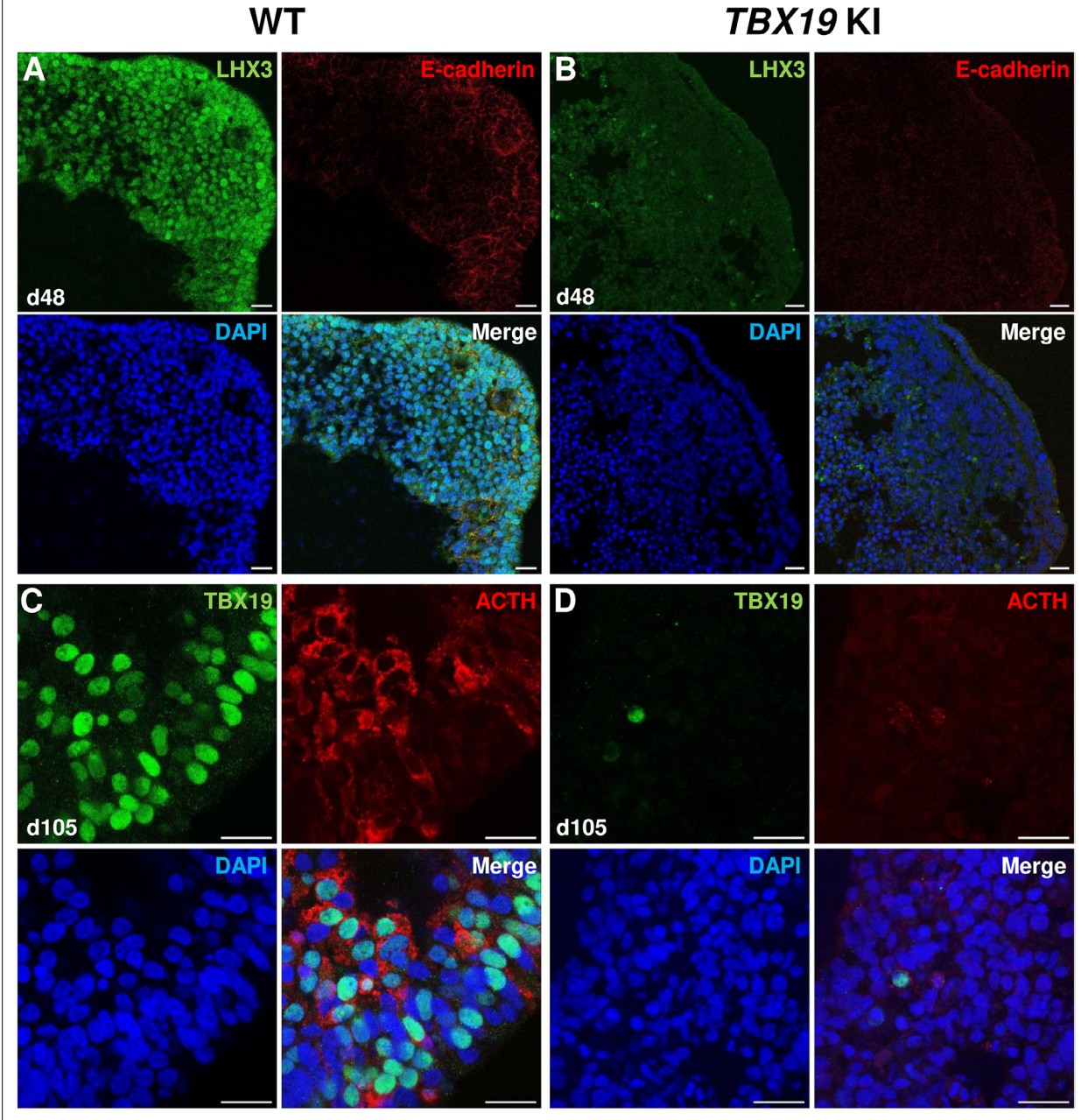

**Figure 3.** Impairment of corticotroph development in *TBX19* KI organoids as compared with controls at the protein level. (**A, B**) Immunostaining of LHX3 and CDH1 (E-cadherin) expression in epithelial cells, typical of Rathke's pouch ectoderm in early pituitary primordia, was reduced in *TBX19* KI organoid *vs* wild-type (WT) on day 48 (n=10 organoids for each group). Scale bars: 10 μm. (**C, D**) Immunostaining showed that adrenocorticotropic hormone (ACTH) and TBX19 expressions were reduced in *TBX19* KI organoid *vs* WT on day 105 (n=10 organoids for each group). Scale bars: 10 μm.

(*Figure 5*) from an isogenic control. The *NFKB2*$^{D865G}$ homozygous mutation was chosen because it has been shown to severely affect NFKB2 p52 processing (*Lee et al., 2014*). Two homozygous KI clones carrying *NFKB2*$^{D865G/D865G}$ were obtained after screening by CAPS and confirmation by Sanger sequencing (*Figure 5—figure supplement 1*). We selected one homozygous missense mutation *NFKB2* KI clone (#7) to amplify and generate 200 mutant organoids in parallel with 200 WT organoids in 3D culture.

Next, we compared the development and differentiation of the *NFKB2* KI organoids versus WT organoids during the culture by qRT-PCR and immunofluorescence as described above. The results of qRT-PCR showed no significant difference in the peak expression level of *HESX1*. However, higher

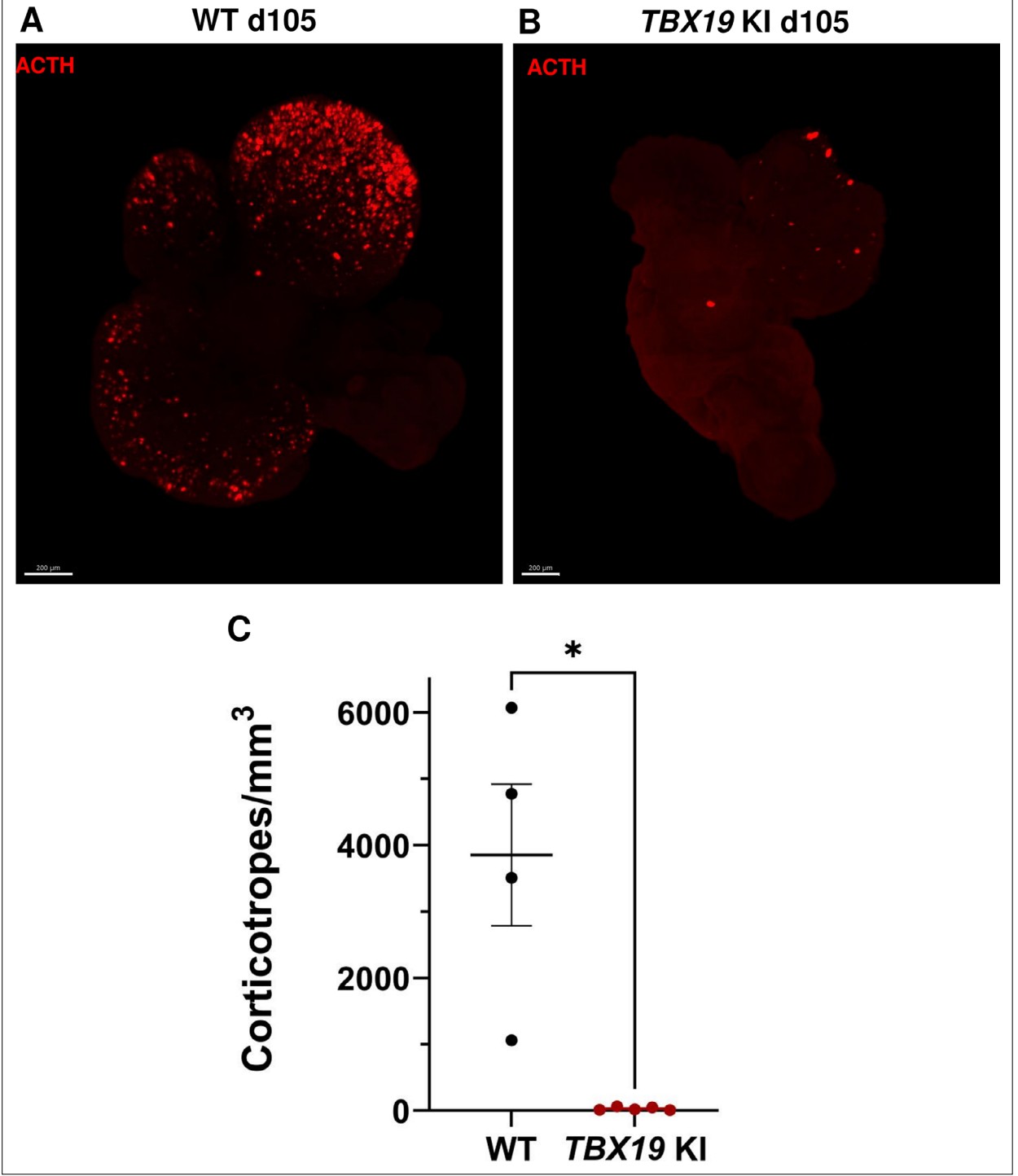

**Figure 4.** 3D reconstruction of whole wild-type (WT) and *TBX19* KI organoids on day 105. (**A**) A representative image of whole-mount immunostaining against adrenocorticotropic hormone (ACTH) in a cleared WT organoid on d105 using light-sheet microscopy. Scale bar: 200 μm. (**B**) A representative image of whole-mount immunostaining against ACTH in a cleared *TBX19* KI organoid on d105 as above, showing impaired corticotroph differentiation. Scale bar: 200 μm. (**C**) The number of corticotroph cells per mm³ was significantly decreased in *TBX19* KI organoids (* p=0.0159). Means ± SEM for n=4 WT, n=5 *TBX19* KI organoids. Mann-Whitney test (unpaired, two-tailed, nonparametric).

The online version of this article includes the following video(s) for figure 4:

**Figure 4—video 1.** 3D animation of whole-mount d105 WT cleared organoid immunostained for TBX19 (green) and adrenocorticotropic hormone (ACTH) (red).

*Figure 4 continued on next page*

*Figure 4 continued*

**Figure 4—video 2.** 3D animation of whole-mount d105 *TBX19* KI cleared organoid immunostained for TBX19 (green) and adrenocorticotropic hormone (ACTH) (red).

https://elifesciences.org/articles/90875/figures#fig4video2

*HESX1* expression in mutant organoids from d27 to d75 suggested impaired downregulation of the gene (*Figure 6B*). Concomitantly, a significantly decreased expression of *PITX1* and *LHX3* in *NFKB2* KI organoids was found from d48 onwards, suggesting reduced differentiation of pituitary progenitors (*Figure 6C–D*). Similar to what we observed in *TBX19* KI mutants, *TBX19* expression was increased in *NFKB2* KI organoids at d105 (*Figure 6E*). However, this had no impact on *POMC* expression levels, which remained significantly lower in *NFKB2 KI* organoids (*Figure 6F*), suggesting a role for NFKB2 in corticotroph terminal differentiation.

As MRI had revealed pituitary hypoplasia in 43% of patients with DAVID syndrome (*Mac et al., 2023*), we investigated whether mutant organoids might be smaller than WT organoids. Comparing the volume of both types of organoids on day 105, we found no significant difference (p=0.6, WT n=7, mutant n=8, *Figure 6G*). In line with qRT-PCR results, immunostaining showed less expression of LHX3 in *NFKB2* KI organoids on day 48 compared to the WT (*Figure 7A–B*). NFKB2 (the antibody did not discriminate between p100/p52 isoforms) was robustly expressed in WT LHX3 + pituitary progenitors on day 48 (*Figure 7A*). Unexpectedly, NFKB2 p100/p52 immunostaining was weaker in *NFKB2* KI organoids than in WT organoids (*Figure 7A–B*). Immunostaining at d105 in *NFKB2* KI organoids showed the presence of many TBX19 + nuclei, in line with the qRT-PCR results, but few of them were surrounded by cytoplasmic ACTH signal (*Figure 7C–D*). Quantification after 3D reconstruction of transparent whole organoids showed significantly fewer ACTH + cells in *NFKB2* KI organoids on day 105 compared to WT organoids in both qualitative (*Figure 8A–B* and *Figure 8—video 1* and *Figure 8—video 2*) and quantitative (*Figure 8C*) assessments. In the absence of an immune system, this is strong evidence in support of a direct role for NFKB2 signaling in pituitary differentiation.

To further investigate other downstream pathways altered in *NFKB2* KI pituitary organoids, we performed whole RNA-seq of five distinct organoids on day 48 from the *NFKB2* KI line versus five organoids derived from its corresponding hiPSC control line. As we found that only 60–70% of organoids show signs of pituitary cell differentiation, we chose to perform a preselection of organoids, based on RT-qPCR expression of selected markers (*SOX2, HESX1, PITX1, LHX3, TBX19, POU1F1,* and *POMC*) in order to avoid having 'empty' HPOs sent for bulk RNA-seq. Differential expression (DE) gene analysis identified 2559 significantly upregulated and 2260 significantly downregulated genes at adjusted p-value (pAdj) <0.05. The global results are depicted as a heatmap to show the similarities across organoids of similar genotypes (*Figure 9A*), and a volcano plot highlights the magnitudes of DE between the two groups (*Figure 9B*, *Figure 9—source data 1*).

NFKB signaling did not seem to be affected in *NFKB2* KI organoids, as most genes involved in that pathway had fold-change values between –0.5 and 0.5 when significant (*Figure 9—figure supplement 1*). Among a list of 144 genes known to have a functional influence on pituitary-hypothalamic development as curated from the published literature, 67 were found to be differentially expressed in *NFKB2* KI organoids (pAdj <0.05) (*Figure 9B*, and *Table 1*), of which 39 encode transcription factors (*Figure 9C*). Expression of these transcription factors was mostly downregulated, with a marked decrease in expression of pituitary progenitor markers such as *PITX1* and *LHX3*. In contrast, the earliest Rathke's pouch marker, *HESX1,* showed a twofold increased expression in *NFKB2* KI organoids, suggesting impaired progression of pituitary progenitors towards more differentiated stages.

During pituitary development, progenitors undergo epithelial-to-mesenchymal transition (EMT). Data from human fetuses (*Zhang et al., 2020*) suggest that pituitary stem cells/progenitors are in a hybrid state, as they simultaneously express 'stemness' as well as epithelial and mesenchymal-associated genes, and expression of all these gene sets decreases in concert during the course of differentiation. While the overall number of anterior pituitary-type cells in *NFKB2* KI organoids was not affected, as confirmed by RT-qPCR of the pan-pituitary cell adhesion marker *EPCAM*, there was a significant decrease in the expression of stemness markers such as *HES1, SOX2,* and *SOX9* (*Table 2*, left panel), accompanied by increased expression of stem cell-associated epithelial markers *CDH1* and *KRT8* (*Table 2*, middle panel) and decreased expression of stem cell-associated mesenchymal markers *CDH2* and *VIM*. Other mesenchymal markers expressed in both stem cells and differentiating cells

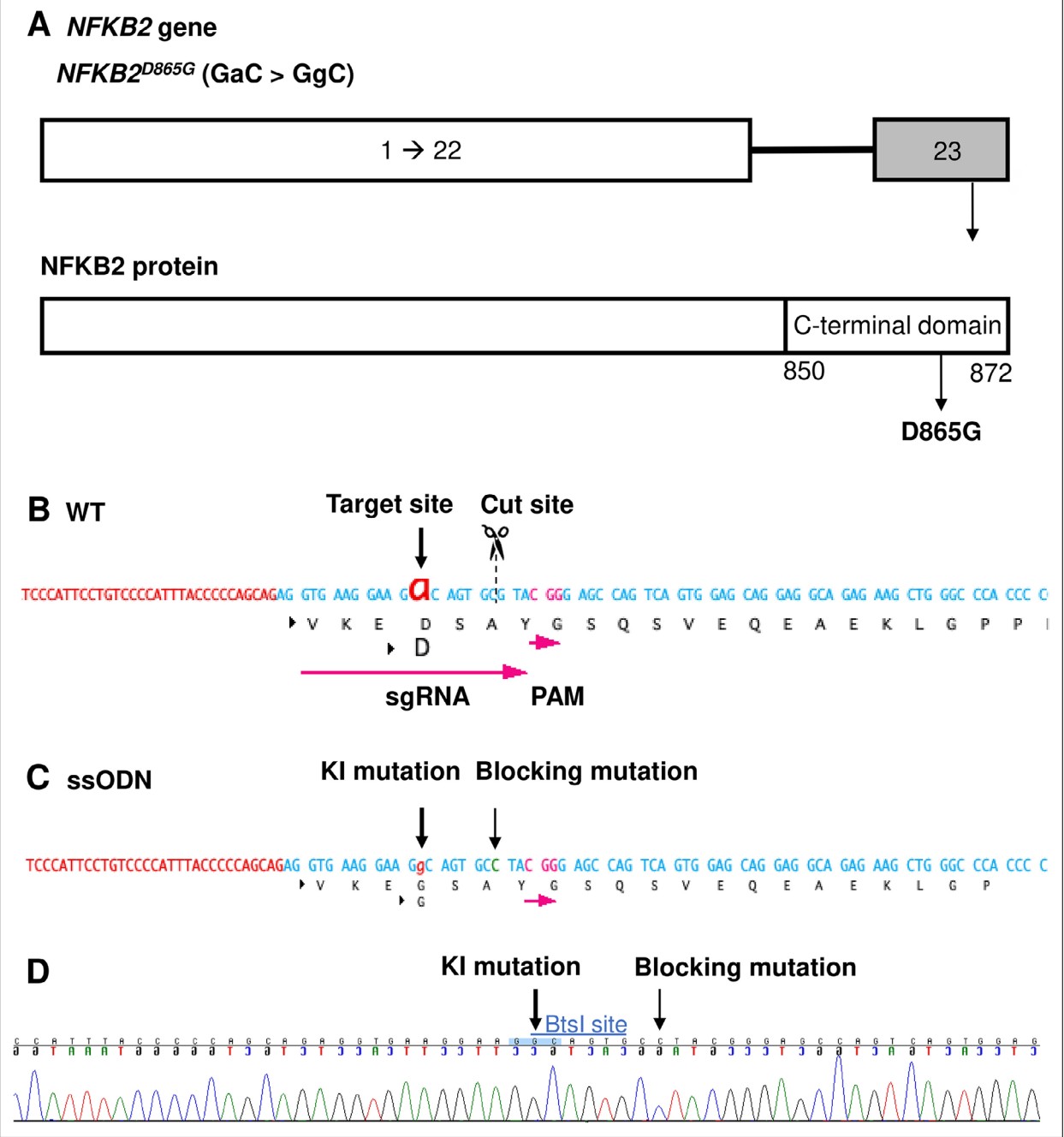

**Figure 5.** Design of sgRNA and single-stranded oligo DNA nucleotides (ssODN) to introduce a nuclear factor kappa-B subunit 2 (*NFKB2*)^D865G mutation. (**A**) Illustration of the *NFKB2* gene (HGNC: 7795; ENSEMBL: ENSG00000077150; Human GRCh38). (**B**) Wild-type (WT) sequence containing target site, cut site, target, and protospacer-adjacent motif (PAM) sequence. (**C**) ssODN design to introduce the missense mutation D865G into *NFKB2* using CRISPR/Cas9. (**D**) Sequence analysis of the *NFKB2* KI hiPSC clone 7, obtained by Sanger sequencing, after screening by cleaved amplified polymorphic sequences (CAPS). This clone was subsequently used in this work to differentiate into pituitary organoids (see below).

The online version of this article includes the following source data and figure supplement(s) for figure 5:

**Figure supplement 1.** Results of cleaved amplified polymorphic sequences (CAPS) assay and Sanger sequencing for editing nuclear factor kappa-B subunit 2 (*NFKB2*) mutation.

**Figure supplement 1—source data 1.** A pdf file describing modifications of original pictures used in *Figure 5—figure supplement 1*.

**Figure supplement 1—source data 2.** Two original pictures used in *Figure 5—figure supplement 1*.

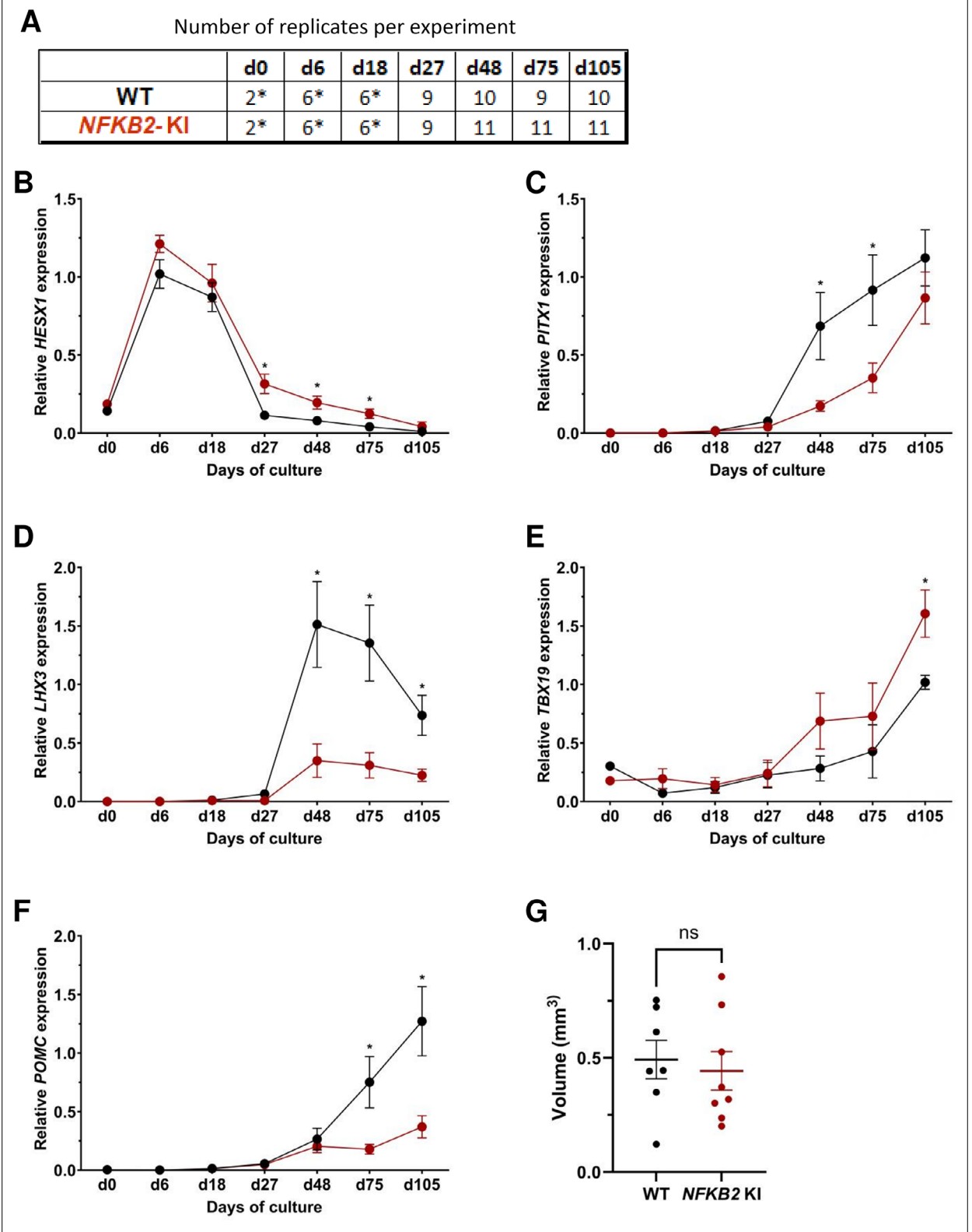

**Figure 6.** Time course of organoid growth and gene expression in wild-type (WT) and nuclear factor kappa-B subunit 2 (*NFKB2*) KI organoids. (**A**) Number of replicates per time point and per genotype in experiments depicted in the following graphs. Asterisks indicate that 7–8 organoids were grouped for each sample. For other point, each sample consists of a single organoid. (**B–F**) Relative quantification (RQ) mRNA expression analysis for key markers of pituitary organoids during differentiation: WT (black line and dots) and *NFKB2* KI organoids (red line and dots). Data show means

*Figure 6 continued on next page*

Figure 6 continued

± standard error of the mean (SEM);. Mann-Whitney t-test (unpaired, two-tailed, nonparametric). p<0.05 (*), p<0.01 (**). (**B**) Relative quantification expression of *HESX1*, the earliest pituitary placode marker assessed. The expression of *HESX1* was upregulated in *NFKB2* KI organoids vs. WT between d27 and d75. (**C**) Pituitary progenitor marker *PITX1* was significantly downregulated in *NFKB2* KI organoids at d48 and d75. (**D**) Pituitary progenitor marker *LHX3* was significantly lower in *NFKB2* KI organoids as compared to WT from d48 onwards. (**E**) Relative quantification expression of *TBX19*, a corticotroph marker. *TBX19* was significantly increased in *NFKB2* KI organoids by d105. (**F**) Relative quantification expression of pro-opiomelanocortin (*POMC*), a corticotroph marker. *POMC* was significantly downregulated in *NFKB2* KI organoids from d75. (**G**) Volume of organoids (mm³) on d105, calculated using Imaris software (see in methods). There was no significant difference in volume between WT and *NFKB2* KI organoids (p=0.6126). Data show means ± SEM; n=7 in the WT group, n=8 in the mutant group. Mann-Whitney test (unpaired, two-tailed).

(*COL1A1* and *COL1A2*) were upregulated (*Table 2*, right panel). *CLDN6*, which is expressed in early pituitary progenitors (*Zhang et al., 2020*), was upregulated (fold change = 2.3; p<0.0001) whereas *CLDN9*, widely expressed in the fetal human pituitary (*Zhang et al., 2020*), was decreased (fold change = 0.34; p<0.0001). Overall, these results suggest that progenitors initiate EMT, but stall during the process. Of note, the expression of *ZEB2*, *SNAI1*, and *SNAI2*, key initiators of EMT through the downregulation of *CDH1*, is significantly increased (fold changes of 1.37, 3.37, and 5.36, respectively; p<0.01), suggesting that *NFKB2* plays an important role downstream from these effectors.

## *TBX19* KI and *NFKB2* KI alter hypothalamic-pituitary organoid development through distinct mechanisms

The transcription factor *RAX,* which is physiologically expressed in the hypothalamus, also showed a moderate but significant decrease in its expression in *NFKB2* KI organoids (*Figure 9C* and *Table 1*), suggesting that the phenotype could at least partially be due to a hypothalamic defect. Indeed, among the growth factors known to mediate the interaction between the hypothalamus and oral ectoderm during pituitary development in animal models, *BMP4, FGF8,* and *FGF10* had fold-changes of 4.35, 0.5, and 0.26 in *NFKB2* KI organoids respectively, whereas *SHH, WNT5A,* and *FGF8* expression levels were not modified (*Table 1*).

In comparing the results of RT-qPCR to those obtained by RNA-seq, we found good correlations (R²=0.7595; p<0.0001) between the *NFKB2* KI to WT ratios measured with both techniques on the same samples, except for *TBX19* (*Figure 9—figure supplement 2*). As we obtained similar results with 2 sets of *TBX19* RT-qPCR primers, targeting different exons, and with at least one primer hybridizing over the junction of two exons, we believe the RNA-seq results for this gene to be likely biased, though we have not identified the reason.

Comparing gene expression levels between *TBX19* KI and *NFKB2* KI organoids of developmentally important transcription and growth factors shows their differences. As mentioned before, *HESX1* was upregulated at d48 only in *NFKB2* KI organoids, whereas both mutants had decreased levels of *PITX1* and *LHX3*, the latter being more severely impacted in *NFKB2* KI organoids (*Figure 10A*). In these, we confirmed the increase in *BMP4* and decrease in *FGF8/FGF10* expression seen in RNA-seq. We did not find any significant differences in these transcripts between WT and *TBX19* KI organoids (*Figure 10B*), suggesting that the impaired development of pituitary progenitors in this model has a different mechanism.

We then explored markers of different stages of corticotroph maturation that were recently identified in human fetal pituitary (*Zhang et al., 2020*). RNA-seq on d48 *NFKB2* KI organoids showed increased expression of *FST*, a gene expressed in immature corticotropes, and concomitant decreased levels of fully mature corticotrope markers *AR*, *NR4A2*, *PCSK1*, and *POMC* (0.63, 0.4, 0.35, and 0.43-fold changes, respectively) (*Table 1*). RT-qPCR at d48, d75, and d105 showed that *TBX19* was upregulated at later stages in both models, suggesting that both *TBX19* and p52 are involved in the control of *TBX19* expression (*Figure 11A*). The *NR4A2* transcription factor was downregulated at d75 in both mutants, but strongly upregulated in *TBX19* KI organoids and downregulated in *NFKB2* KI organoids at d105 (*Figure 11B*). *PCSK1* expression, necessary for prohormone processing, also decreased in both mutants, but this phenomenon started earlier in *NFKB2* KI organoids (*Figure 11C*). Finally, we found that *POMC* expression was as strongly impaired in *NFKB2* KI organoids as in *TBX19* KI organoids. These results, combined with normal expression levels for *NEUROD1*, corroborate the hypothesis that TBX19-positive precursors fail to achieve terminal differentiation in *NFKB2* KI organoids.

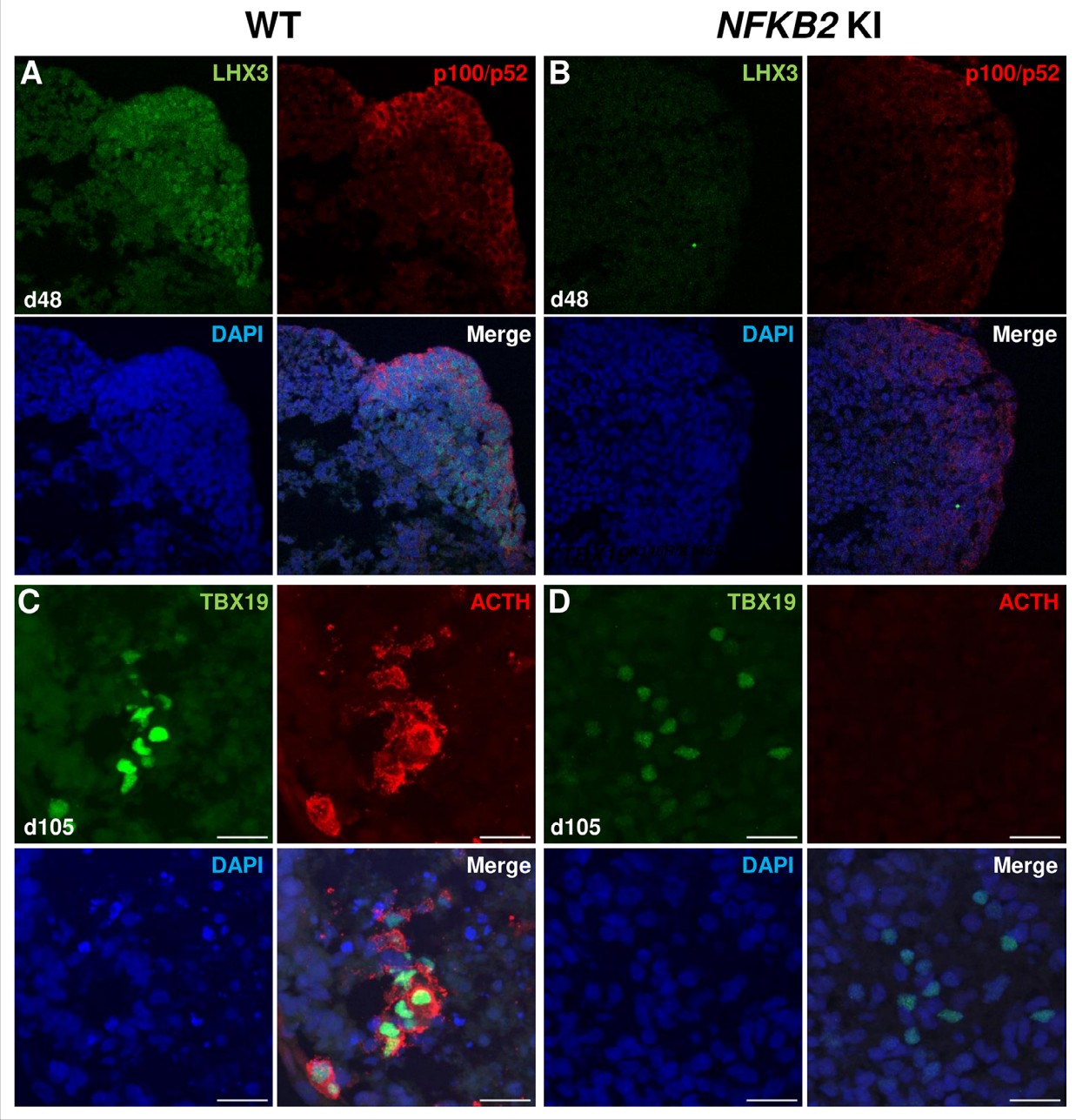

**Figure 7.** Impairment of pituitary development in nuclear factor kappa-B subunit 2 (*NFKB2*) KI organoids. (**A, B**) Immunostaining for LHX3 (green) and p100/p52 (red) in the early pituitary-type epithelium was reduced in *NFKB2* KI organoids vs wild-type (WT) on day 48 (n=10 organoids for each group). Stronger expression of p100/p52 was observed in pituitary progenitors, but expression in the hypothalamic part of the organoid cannot be excluded. Scale bar: 10 μm (**C, D**) by day 105, although nuclear TBX19 was detectable in both mutants, these cells failed to co-express ACTH in *NFKB2* KI organoids (n=10 organoids for each group). Scale bars: 10 μm.

Interestingly, other lineages also appeared to be affected, both in *TBX19* KI and *NFKB2* KI organoids. RNA-seq at d48 showed drastic downregulation of *PROP1* and *POU1F1* transcription factors in *NFKB2* mutants. Although we could not accurately measure *POU1F1* expression at d48 due to its too low expression, RT-qPCR (*Figure 12*) showed that *PROP1* downregulation was also occurring in *TBX19* KI organoids at d48 (*Figure 12A*, left), and both *PROP1* and *POU1F1* levels were barely detectable at d75 in the two models (*Figure 12A and D*). This is likely to be the consequence of the impaired generation of pituitary progenitors described above. Nonetheless, other transcription factor markers of *POU1F1*-dependent lineages were expressed, as shown by normal expression of *NEUROD4* and

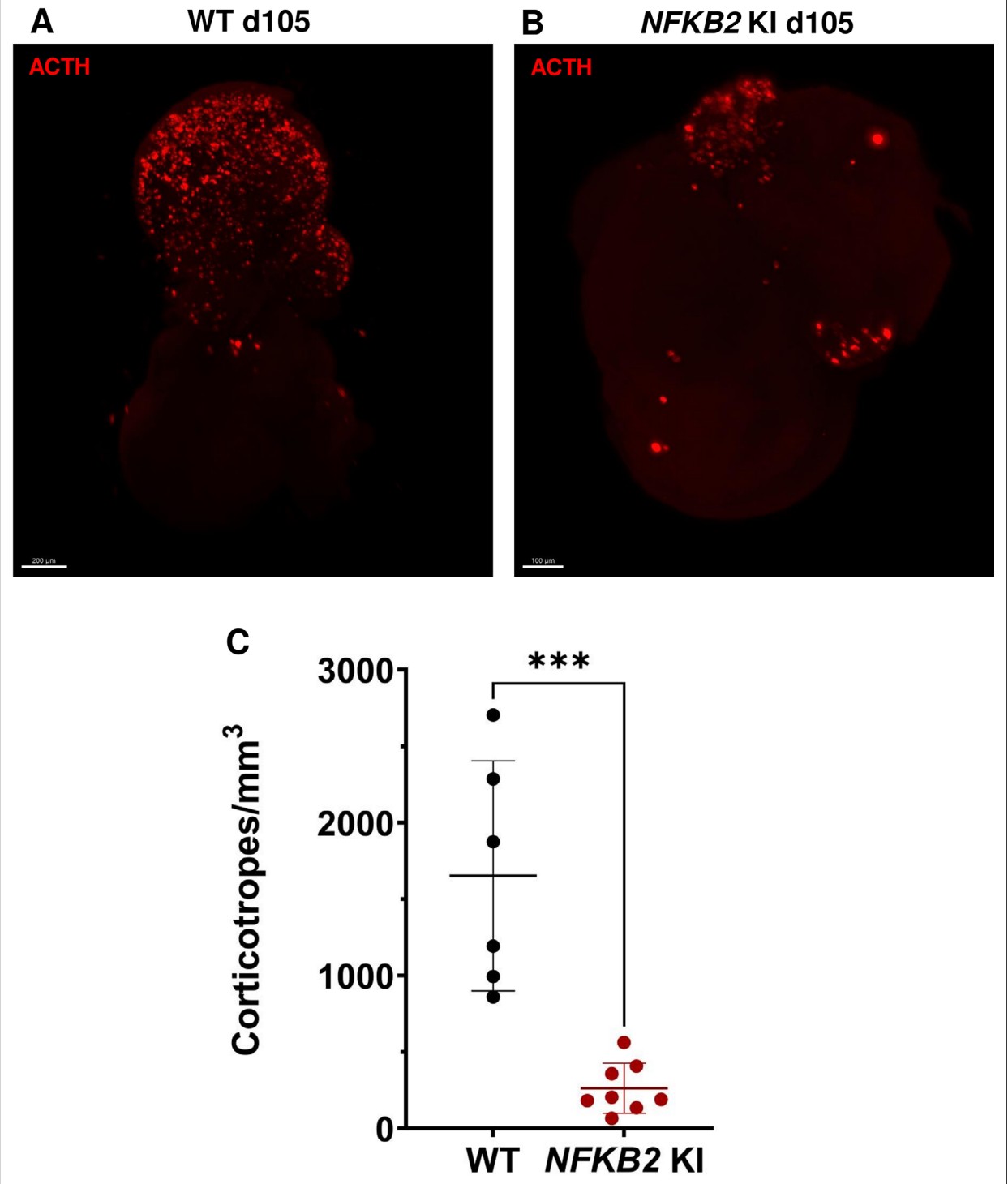

**Figure 8.** 3D reconstruction of whole wild-type (WT) and nuclear factor kappa-B subunit 2 (*NFKB2*) KI organoids on day 105. (**A**) A representative image of whole-mount immunostaining against adrenocorticotropic hormone (ACTH) in a cleared WT organoid on d105 using light-sheet microscopy. Scale bar: 200 μm. (**B**) A representative image of whole-mount immunostaining against ACTH in a cleared *NFKB2* KI organoid on d105 as above, showing impaired corticotroph differentiation. Scale bar: 100 μm. (**C**) The number of corticotroph cells per mm³ was significantly decreased in *NFKB2* KI organoids (*p=0.0007). Means ± SEM for n=6 organoids WT, n=8 *NFKB2* KI organoids. Mann-Whitney test (unpaired, two-tailed, nonparametric).

The online version of this article includes the following video(s) for figure 8:

**Figure 8—video 1.** 3D animation of whole-mount d105 wild-type (WT) cleared organoid immunostained for TBX19 (green) and adrenocorticotropic hormone (ACTH) (red).

*Figure 8 continued on next page*

*Figure 8 continued*

https://elifesciences.org/articles/90875/figures#fig8video1

**Figure 8—video 2.** 3D animation of whole-mount d105 nuclear factor kappa-B subunit 2 (*NFKB2*) KI cleared organoid immunostained for TBX19 (green) and adrenocorticotropic hormone (ACTH) (red).

https://elifesciences.org/articles/90875/figures#fig8video2

---

*ZBTB20* in *TBX19* KI organoids at d48 (*Figure 12B and C*, left). However, both genes had lower expression levels at 75 compared to WT (*Figure 12B and C*, right). In *NFKB2* KI organoids, *NEUROD4* was downregulated at both time points (*Figure 12B*), and *ZBTB20* at d48 only (*Figure 12C*, left), with signs of recovery at d75 (*Figure 12C*, right) and d105 (data not shown). Again, these results suggest that different mechanisms converge on impaired POU1F1-dependent lineage development.

Finally, the glycoprotein hormone common alpha subunit *CGA* was strongly upregulated at d48 (*Table 1*) and d75 (data not shown) in *NFKB2* KI organoids. The concomitant increase in *GATA2*, *CYP11A1*, and *FOLR1* may suggest a redirection of some progenitors towards a gonadotroph/thyrotroph cell fate. However, the lack of changes in the expression of *NR5A1*, *NEUROD1*, and *SOX11*, and the undetectable transcripts for the specific beta subunits *FSHB*, *LHB*, and *TSHB* before d105, do not favour this hypothesis.

A few years ago, ChIP-seq experiments on Hodgkin lymphoma cells (*de Oliveira et al., 2016*) identified 10,893 DNA-binding regions for NFKB2 p52, within or in the close vicinity of 5,497 genes. About half of the p52 recruitments occurred in transcribed regions, with a high percentage within introns, and the other half in intergenic regions close to a transcriptional start site (TSS). Among the 4819 differentially regulated genes identified in *NFKB2* KI organoids at d48, 1398 of them had p52 binding regions nearby (*Figure 13*), of which 23 belong to our curated list of pituitary genes of interest. Among them, we found genes crucial for progenitor development (*SOX9*, *PITX1*), EMT (*CDH1*, *ZEB2*), POU1F1-dependent lineages (*PROP1*, *NEUROD4*), and most importantly, the principal genes of corticotroph development and maturation (*TBX19*, *NR4A2*, *PCSK1*, *POMC*). Thus, these changes in the expression of these genes we observed in *NFKB2* KI organoids may be due to a direct effect on transcriptional regulation.

Altogether, our data identify the NFKB2 signaling pathway as an important factor acting at multiple levels on human pituitary development.

## Discussion

After the description of DAVID syndrome as a rare combination of anterior pituitary deficit, mostly ACTHD, with common variable immunodeficiency, we and others found that it is associated with *NFKB2* gene mutations affecting specific C-terminal residues of the NFKB2 protein, known to play important roles in immunity and also expressed in the pituitary (*Quentien et al., 2012*; *Chen et al., 2013*; *Brue et al., 2014*). As the mouse model harboring a similar alteration of the ortholog gene lacked an endocrine phenotype (*Brue et al., 2014*), we wanted to investigate the mechanism of ACTHD in this syndrome. Currently, available in vitro models to study ACTHD and CPHD have strong limits. For instance, studies based on transfections and luciferase-coupled hormone promoters can only give indirect evidence of the effect of a variant of unknown significance as they are based on partly arbitrary amounts of DNA and promoters in an artificial context (*Jullien et al., 2019*). Murine models only partially replicate human CPHD, as shown for *PROP1* mutations and the occurrence of ACTHD (absent in mice but present in 40% of humans) (*Castinetti et al., 2016*). In the past decade, the reprogramming of differentiated adult cells into induced pluripotent stem cells and advancements in genome editing technologies using CRISPR/Cas9 systems have broadened possibilities for modeling novel aspects of human disease ex vivo (*Randolph et al., 2017*). Organoid culture has become a powerful tool for modeling otherwise inaccessible congenital human disorders in many organs such as brain (*Arnaud et al., 2022*), intestine (*Schwank et al., 2013*), kidney (*Takasato et al., 2015*), liver (*Guan et al., 2017*), pancreas (*Moreira et al., 2018*), ovary (*Nanki et al., 2020*), and lung (*Kong et al., 2021*). In the present study, we established a human in vitro model of congenital ACTHD using 3D pituitary organoids differentiated from *TBX19* KI and *NFKB2* KI hiPSC generated by CRISPR/Cas9 genome editing, in order to compare them with their WT equivalents on the same genetic background. After confirming that our model could replicate pituitary development, we then determined

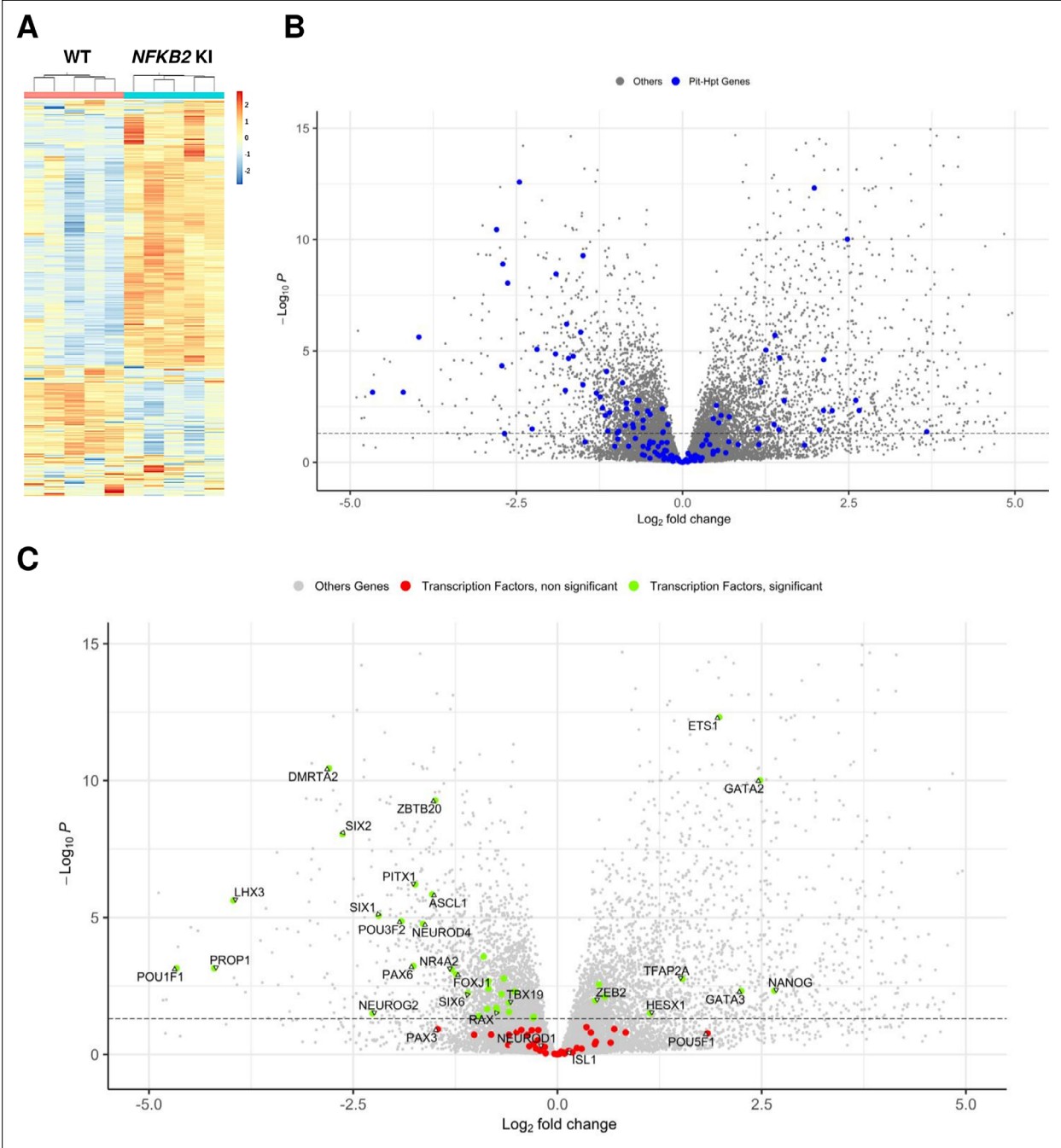

**Figure 9.** Whole-transcriptome profiles of wild-type (WT) vs nuclear factor kappa-B subunit 2 (*NFKB2*) knock-in (KI) organoids on day 48. (**A**) Heat map showing global differential gene expression between WT vs *NFKB2* KI organoids (n=5 for each group). (**B**) Volcano plot of genes showing differential gene expression between WT vs *NFKB2* KI organoids (expressed in Log$_2$ fold-change). Blue dots in the left panel indicate 144 genes involved in hypothalamic-pituitary development. Gray dots indicate all other genes detected in RNA-seq. (**C**) Volcano plot of genes showing differential gene expression between WT vs *NFKB2* KI organoids. Green dots indicate genes coding for 39 key transcription factors whose expression is significantly changed (padj <0.05). Red dots indicate key transcription factors with non-significant changes (padj >0.05). Gray dots indicate all other genes detected in RNA-seq.

The online version of this article includes the following source data and figure supplement(s) for figure 9:

**Source data 1.** Nuclear factor kappa-B subunit 2 *(NFKB2)* nock-in (KI) vs wild-type (WT) gene expression at d48 Excel spreadsheet showing *NFKB2* KI vs WT expression ratios measured by RNA-seq (n=5 organoids of each genotype).

**Figure supplement 1.** Volcano plot of genes showing differential gene expression between wild-type (WT) vs nuclear factor kappa-B subunit 2 (*NFKB2*)$^{D865G/D865G}$ organoids for genes involved in nuclear factor kappa B (NFKB) canonical and non-canonical pathways.

**Figure supplement 2.** Correlation analysis of quantitative real-time PCR (qRT-PCR) and RNA-seq on same the samples.

**Table 1.** RNA-seq expression data for a list of 144 genes known from the literature to have a functional influence on pituitary-hypothalamic development.

Differentially expressed genes (padj <0.05) in nuclear factor kappa-B subunit 2 (*NFKB2*) KI organoids are highlighted in green when upregulated and in orange when downregulated. FC; Fold change.

| Gene name | FC | padj | Gene name | FC | padj | Gene name | FC | padj |
|---|---|---|---|---|---|---|---|---|
| ADCYAP1 | 0,447 | 0,00789 | GLI3 | 1,279 | 0,10094 | PITX2 | 1,072 | 0,86688 |
| AIP | 0,898 | 0,53442 | GNAS | 1,058 | 0,39519 | POLR3A | 1,014 | 0,90718 |
| ALDH1A1 | 0,153 | 1,3E-09 | GNRHR | 0,453 | 8,3E-05 | POMC | 0,434 | 0,0036 |
| ALDH1A2 | 0,152 | 4,6E-05 | GNRHR2 | 0,698 | 0,23938 | POU1F1 | 0,039 | 0,00072 |
| ALDH1A3 | 0,182 | 2,6E-13 | GPR101 | 1,209 | 0,83045 | POU3F1 | 1,374 | 0,42093 |
| AR | 0,634 | 0,00168 | GREB1 | 1,007 | 0,97784 | POU3F2 | 0,266 | 1,4E-05 |
| ARNT2 | 0,534 | 0,00027 | GSX1 | 0,659 | 0,45012 | POU5F1 | 3,573 | 0,17018 |
| ASCL1 | 0,345 | 1,4E-06 | HES1 | 0,663 | 0,02805 | PROKR2 | 1,056 | 0,97365 |
| AXIN2 | 1,296 | 0,05889 | HESX1 | 2,193 | 0,03121 | PROP1 | 0,054 | 0,00071 |
| BMP2 | 0,750 | 0,34278 | HEY1 | 0,804 | 0,13046 | RAX | 0,599 | 0,02697 |
| BMP4 | 4,357 | 2,4E-05 | HMGA2 | 0,775 | 0,19471 | RBPJ | 0,814 | 0,04665 |
| BMPR1A | 1,019 | 0,91991 | INHBB | 0,458 | 0,03974 | S100B | 0,304 | 2,2E-05 |
| BRAF | 0,892 | 0,48183 | ISL1 | 1,097 | 0,75812 | SALL1 | 1,106 | 0,76122 |
| CDH1 | 1,626 | 0,009 | KRT8 | 2,617 | 2E-06 | SHH | 0,786 | 0,51179 |
| CGA | 12,756 | 0,04234 | LATS1 | 0,857 | 0,01997 | SIX1 | 0,219 | 8,4E-06 |
| CREB1 | 0,850 | 0,12901 | LATS2 | 1,460 | 0,01663 | SIX2 | 0,161 | 9E-09 |
| CRH | 4,347 | 0,00469 | LHX2 | 0,622 | 0,00625 | SIX3 | 0,859 | 0,69976 |
| CRHR1 | 0,156 | 0,05131 | LHX3 | 0,064 | 2,4E-06 | SIX6 | 0,467 | 0,00586 |
| CRHR2 | 0,823 | 0,73602 | LHX4 | 0,557 | 0,00398 | SLC15A2 | 0,624 | 0,00164 |
| CTNNB1 | 1,009 | 0,95542 | MEN1 | 1,030 | 0,80309 | SLC6A3 | 2,215 | 0,15797 |
| CYP11A1 | 6,086 | 0,00166 | MSX1 | 0,904 | 0,90909 | SMO | 0,957 | 0,70401 |
| DIO2 | 1,114 | 0,71885 | NANOG | 6,301 | 0,00472 | SMOC2 | 4,163 | 0,03488 |
| DISP1 | 1,142 | 0,47708 | NEUROD1 | 0,883 | 0,50966 | SOX11 | 0,736 | 0,12757 |
| DLK1 | 0,594 | 0,01988 | NEUROD2 | 1,137 | 0,84178 | SOX2 | 0,815 | 0,42151 |
| DMRTA2 | 0,144 | 3,6E-11 | NEUROD4 | 0,320 | 1,7E-05 | SOX3 | 0,787 | 0,50152 |
| DRD2 | 2,753 | 2E-05 | NEUROG1 | 0,493 | 0,19209 | SOX9 | 0,557 | 0,00214 |
| DUOX2 | 2,595 | 0,01961 | NEUROG2 | 0,208 | 0,03165 | SRD5A1 | 0,961 | 0,79794 |
| DZIP1 | 0,818 | 0,04264 | NFKB2 | 1,236 | 0,16144 | SST | 0,994 | 0,9921 |
| EGR1 | 1,784 | 0,15768 | NKX2-1 | 0,707 | 0,15518 | SSTR1 | 0,509 | 0,09008 |
| EPCAM | 1,128 | 0,72728 | NKX2-2 | 1,569 | 0,37322 | SSTR2 | 0,769 | 0,43599 |
| ETS1 | 3,947 | 4,8E-13 | NOG | 1,376 | 0,31897 | TAZ | 0,967 | 0,85318 |
| EYA1 | 0,514 | 0,03962 | NOTCH1 | 1,006 | 0,98289 | TBX19 | 0,663 | 0,01294 |
| FGF10 | 0,267 | 3,5E-09 | NOTCH2 | 0,937 | 0,60691 | TBX3 | 0,997 | 0,98911 |
| FGF2 | 1,224 | 0,18237 | NR0B1 | 1,062 | 0,92893 | TCF4 | 1,015 | 0,95769 |
| FGF3 | 0,611 | 0,08445 | NR3C1 | 0,846 | 0,30279 | TCF7L2 | 1,178 | 0,58844 |
| FGF8 | 0,506 | 0,04827 | NR4A1 | 1,227 | 0,62126 | TFAP2A | 2,887 | 0,00173 |

*Table 1 continued on next page*

*Table 1 continued*

| Gene name | FC | padj | Gene name | FC | padj | Gene name | FC | padj |
|---|---|---|---|---|---|---|---|---|
| *FOLR1* | 2,729 | 0,03443 | *NR4A2* | 0,407 | 0,00078 | *TGIF1* | 1,422 | 0,00278 |
| *FOXA1* | 0,550 | 0,02193 | NR5A1 | 0,834 | 0,61396 | *THRA* | 0,714 | 0,00712 |
| FOXC1 | 1,619 | 0,11818 | NUPR1 | 1,384 | 0,3477 | *THRB* | 2,260 | 0,00025 |
| *FOXJ1* | 0,424 | 0,00117 | OLFM1 | 0,830 | 0,29166 | *TIPRL* | 0,810 | 0,00393 |
| *FOXO1* | 1,494 | 0,00782 | OTP | 0,663 | 0,19349 | *TLE5* | 0,695 | 0,00502 |
| *FST* | 2,384 | 9,1E-06 | OTX1 | 0,863 | 0,72226 | TSHR | 0,998 | 0,99753 |
| *GATA2* | 5,579 | 9,7E-11 | OTX2 | 0,976 | 0,94713 | USP39 | 1,035 | 0,7908 |
| *GATA3* | 4,758 | 0,00483 | PAX3 | 0,362 | 0,11963 | WNT4 | 0,712 | 0,11665 |
| GHRH | 0,712 | 0,65053 | *PAX6* | 0,294 | 0,00059 | WNT5A | 1,442 | 0,29543 |
| GHRHR | 0,672 | 0,48512 | PAX7 | 0,570 | 0,1882 | YAP1 | 1,091 | 0,56585 |
| GLI1 | 1,050 | 0,84104 | *PCSK1* | 0,354 | 0,00033 | *ZBTB20* | 0,354 | 5,3E-10 |
| GLI2 | 1,328 | 0,1593 | *PITX1* | 0,299 | 6,3E-07 | *ZEB2* | 1,378 | 0,01078 |

whether it could also replicate a disease, namely ACTHD. As the *TBX19*[K146R] homozygous variant led to ACTHD in humans, we designed a human 'KI' organoid model derived from *TBX19*[K146R/K146R] hiPSC. In line with the *Tbx19*[-/-] mouse model (*Pulichino et al., 2003*), *TBX19* KI organoids displayed markedly defective corticotroph differentiation, as compared to WT organoids, thereby validating the methodological approach. Interestingly, in organoids carrying a *TBX19*[K146R/K146R] missense mutation that affects DNA binding, a few corticotrophs were still present, suggesting either a role for TBX19 independent of its DNA binding activity or persistence of residual DNA binding activity with this mutation. This model also raises the possibility for a previously undescribed role of TBX19 during early development and the generation of pituitary progenitors in human. However, as *TBX19* mutations have only rarely been associated with other pituitary deficits - especially transient GHD (*McEachern et al., 2011*), further investigations will be needed.

Using this same approach with *NFKB2*[D865G/D865G] human organoids generated by CRISPR/Cas9 genome edited hiPSC, we demonstrated for the first time a role for NFKB2 in pituitary development. Heterozygous mutations in the C-terminal region of *NFKB2*, such as D865G, were reported in DAVID syndrome, leading to the disruption of both non-canonical and canonical pathways (*Chen et al., 2013*; *Kuehn et al., 2017*; *Lee et al., 2014*; *Lindsley et al., 2014*; *Liu et al., 2014*; *Maccari et al., 2017*). Even though *NFKB2*[D865G] was identified in a heterozygous state in patients with ACTHD, we decided to use a homozygous model of *NFKB2* KI to determine whether NFKB2 was involved in pituitary development and could explain the defect in ACTH production in DAVID syndrome (OMIM#, 615577) (*Quentien et al., 2012*; *Chen et al., 2013*; *Mac et al., 2023*). Indeed, while concomitant common variable immune deficiency (CVID) can be explained by the key roles of NFKB signaling in the immune

**Table 2.** Differential expression analysis for genes associated with different stages of epithelial to mesenchymal transition in nuclear factor kappa-B subunit 2 (*NFKB2*)[D865G/D865G] vs wild-type (WT) organoids.
Significantly decreased and increased expressions are highlighted in orange and green, respectively. FC, Fold change.

| Stemness markers | | | Epithelial markers | | | Mesenchymal markers | | |
|---|---|---|---|---|---|---|---|---|
| Gene name | FC | padj | Gene name | FC | padj | Gene name | FC | padj |
| SOX2 | 0,815 | 0,4215 | CDH1 | 1,626 | 0,0090 | CDH2 | 0,616 | 0,0003 |
| SOX4 | 0,784 | 0,0245 | EPCAM | 1,128 | 0,7273 | SNAI1 | 3,379 | 7,14E-06 |
| SOX9 | 0,557 | 0,0021 | KRT8 | 2,617 | 1,98E-06 | VIM | 0,663 | 0,0366 |
| NOTCH2 | 0,937 | 0,6069 | CLDN4 | 1,203 | 0,3140 | COL2A1 | 1,662 | 0,0849 |
| HES1 | 0,663 | 0,0280 | GRHL2 | 1,360 | 0,1920 | COL1A1 | 10,939 | 7,76E-06 |

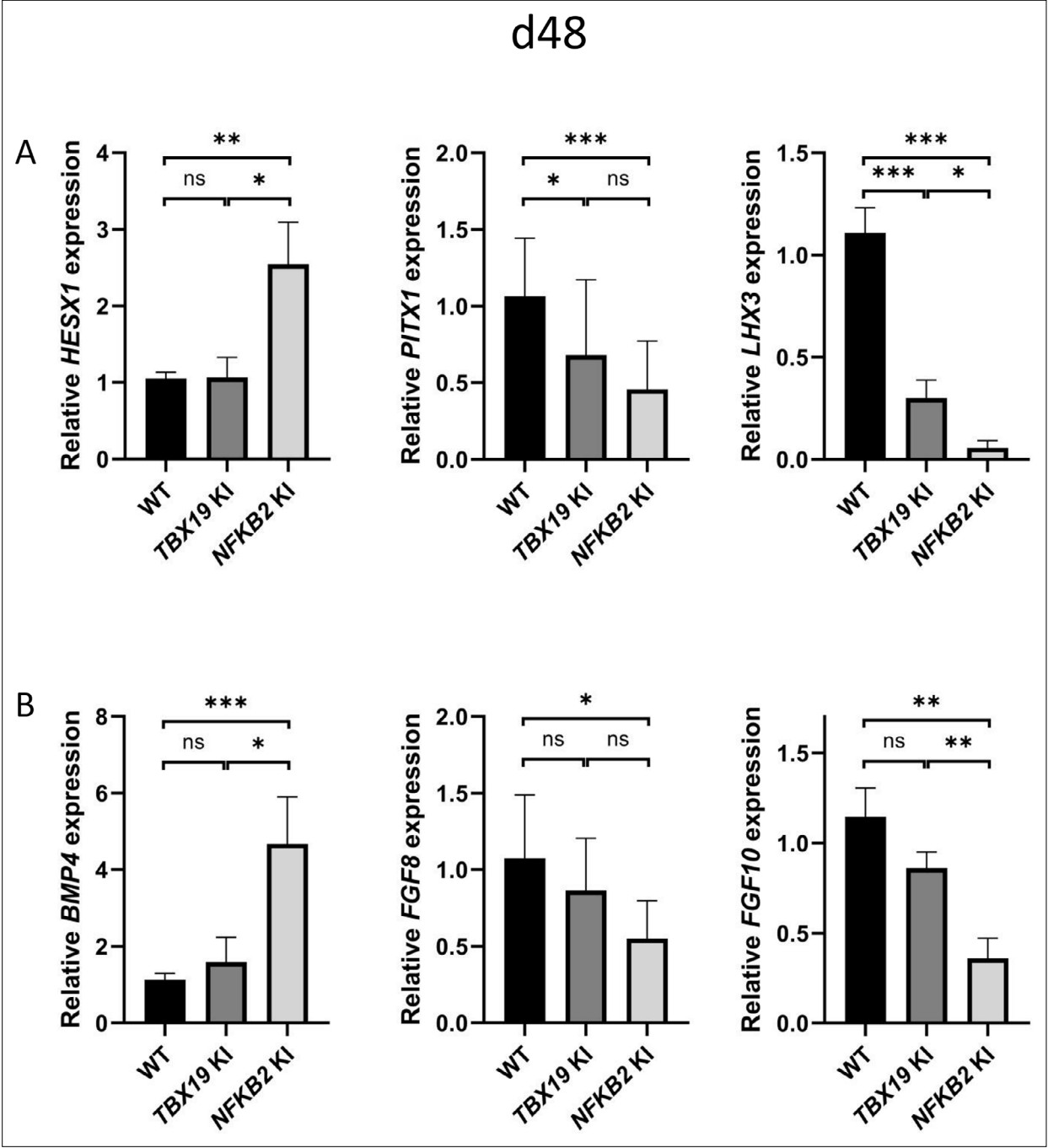

**Figure 10.** Impaired pituitary progenitor development is correlated with changes in growth factor expression in nuclear factor kappa-B subunit 2 (*NFKB2*) knock-in (KI) organoids. (**A**) RT-qPCR experiments at d48 show that *HESX1* expression is upregulated *NFKB2* KI organoids, compared to wild-type (WT) and *TBX19* KI, whereas *PITX1* and *LHX3* are downregulated in both mutants. (**B**) *BMP4* expression is increased whereas, and *FGF8* and *FGF10* are decreased in *NFKB2* mutants, but unchanged in *TBX19* KI organoids. Data show means ± *SEM*; n=17, 12, and 11 for WT, *TBX19* KI, and *NFKB2* KI, respectively. Mann-Whitney t-test (unpaired, two-tailed, nonparametric). p<0.05 (*), p<0.01 (**), p<0.005 (***),.

system, the mechanism of the endocrine deficits caused by *NFKB2* mutants was unknown. In mouse models, deletion of *Nfkb2* causes abnormal germinal center B-cell formation and differentiation (*Carragher et al., 2004*). However, the *Lym1* mouse model, which carries a truncating variant Y868* in the *Nfkb2* orthologue that prevents its cleavage into a transcriptionally active form, had apparently normal corticotroph function and ACTH expression in the pituitary in both heterozygotes and homozygotes (*Brue et al., 2014*), despite reduced fertility and more severe immune abnormalities

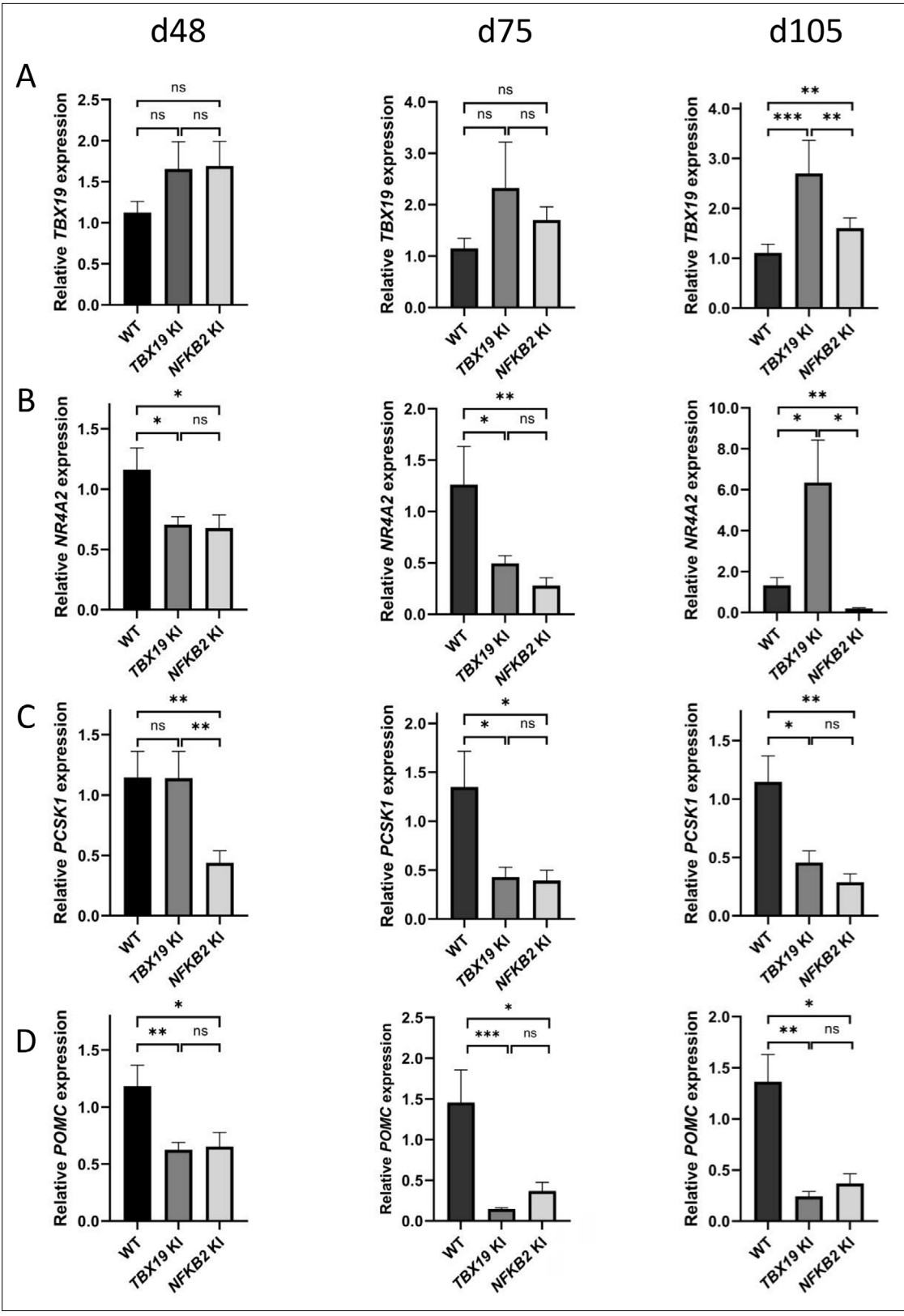

**Figure 11.** Impaired differentiation of corticotrophs in *TBX19* KI and nuclear factor kappa-B subunit 2 (*NFKB2*) knock-in (KI) organoids. (**A**) RT-qPCR measurements of *TBX19* expression show a late (d105) increase in both mutants, with a stronger effect in *TBX19* KI organoids. (**B**) Corticotroph terminal differentiation marker *NR4A2* was decreased in both mutants at d48 and d75, but only in *NFKB2* mutants at d105. (**C**) *PCSK1* downregulation is only observed in *NFKB2* KI at d48, and in both models at d75 and d102. (**D**) A decrease in pro-opiomelanocortin (*POMC*) expression is observed in both

*Figure 11 continued on next page*

Figure 11 continued

mutants from d48 onwards. Data show means ± SEM; d48: n=17, 12, and 11; d75, n=9, 7, and 6; d105: n=9, 5, and 5 for WT, *TBX19* KI, and *NFKB2* KI organoids, respectively. Mann-Whitney t-test (unpaired, two-tailed, nonparametric). p<0.05 (*), p<0.01 (**), p<0.005 (***).

in mutant *NFKB2* homozygotes (*Tucker et al., 2007*). Using pituitary organoids differentiated from a *NFKB2* KI hiPSC line in comparison with its unmutated control and in the absence of an immune response, we have demonstrated that human corticotroph development is directly disrupted by the missense mutation found in DAVID syndrome. The absence of global alteration of the NFKB signaling pathway shows that mutant p100/p52 protein is directly responsible for the observed phenotype.

The precise molecular mechanisms by which *NFKB2* mutation impairs corticotroph development remain to be determined. However, our results (summarized in *Figure 14*) suggest both early effects on the maturation of pituitary progenitors, on epithelial to mesenchymal transition, and late effects on corticotroph differentiation. First, The *NFKB2* gene product p52 could be involved in the early steps of pituitary development. RNA-seq data on day 48 of *NFKB2* KI organoid development showed increased *HESX1* and *BMP4* transcription, in favor of a blockade in an undifferentiated state evocative of the Rathke's pouch epithelial primordium: indeed, down-regulation of HESX1 is necessary for proper pituitary development (*Cohen et al., 2003*). This is also suggested by RNA-seq results showing downregulation of *PITX1*, *HES1*, *LHX3*, and *FGF10* in mutant organoids, whereas their expression is necessary for progression to a progenitor state (*Ohuchi et al., 2000*; *Ericson et al., 1998*). The shift in the *BMP4/FGF10* balance favors the idea that the impaired differentiation of pituitary progenitors is at least in part of hypothalamic origin. Our results also suggest a disorganized EMT process, and the need for a functional p100/p52 to mediate the action of EMT inducers on CDH1 downregulation. Finally, concordant with the hypothesis of a differentiation blockade, upregulation of epithelial markers such as *CDH1* and *KRT8* was observed in mutant *NFKB2* KI organoids on day 48. In contrast, RNA-seq showed no change in Wnt and Hedgehog signaling pathways in *NFKB2* KI organoids.

The *NFKB2* gene product p52 could also be involved in the final steps of corticotroph differentiation. While qRT-PCR showed increased levels of *TBX19* mRNA in mutant organoids, this did not lead to the expected increase in *POMC* mRNA transcripts. Moreover, some *NFKB2* mutants had TBX19 + cells without ACTH expression. This suggests that p52 is necessary for POMC synthesis, which *NFKB2*^D865G may prevent. This could be due to an abnormal DNA binding of the NFKB heterodimer to the *POMC* promoter (*Giraldi et al., 2011*; *Drouin, 2016*), but as other NFKB binding sites are found in intronic regions, it may also be involved in changes in chromatin structures required for proper transcription. The same possibilities apply to *PCSK1*, the gene coding for the enzyme necessary for POMC to ACTH conversion, and to other genes involved in corticotroph maturation. Besides, *NFKB2*^D865G could also block TBX19 + cells in a pre-differentiated state of corticotroph cells expressing FST through its derepression of chromatin remodeling factors such as EZH2 (*De Donatis et al., 2016*).

GH deficiency has been reported in some patients with heterozygous *NFKB2* mutations (*Mac et al., 2023*). The drastic effect on *POU1F1* expression we observed is likely to be the consequence of different phenomena. First, the impaired development of progenitors, and the disruption of *PROP1* expression, possibly due to the absence of p52. Second, both *FOXO1* and *NEUROD4* have p52 binding regions and were differentially expressed. Alongside *POU1F1*, these two genes have been described as crucial members of a feedforward loop controlling somatotroph function (*Stallings et al., 2024*). The 12-fold increase in *CGA* expression and increased expression of pre-gonadotropes or mature thyrotropes (*GATA2*) and mature gonadotropes (*GATA2, CYP11A1,* and *FOLR1*) could be an argument in favor of a redirection of pituitary progenitors towards these lineages. However, the lack of differential expression of *NR5A1* contradicts this hypothesis. A more likely explanation is that *CGA* repression by TBX19 may directly require p52.

The organoid model described in this article presents several advantages over other approaches. It is a true human model of ACTHD, emphasizing the limits of mouse models, as exemplified by the *Lym1* model, which showed that *Nfkb2* p100/p52 function was not essential for pituitary development in mice, but did not provide insights for humans. *Nfkb2*^Lym1/Lym1 mice and our *NFKB2* KI model have different but functionally very similar mutations, as they both lead to an abnormal processing of p100 with its accumulation leading to a strong reduction of p52 content. The phenotype of *Nfkb2*^Lym1/Lym1 being more severe than in mice lacking the complete NFKB2 protein (*Nfkb2*^-/-), these dominant-negative mutations have been called 'super repressors' (*Tucker et al., 2007*). In light of our results, it is

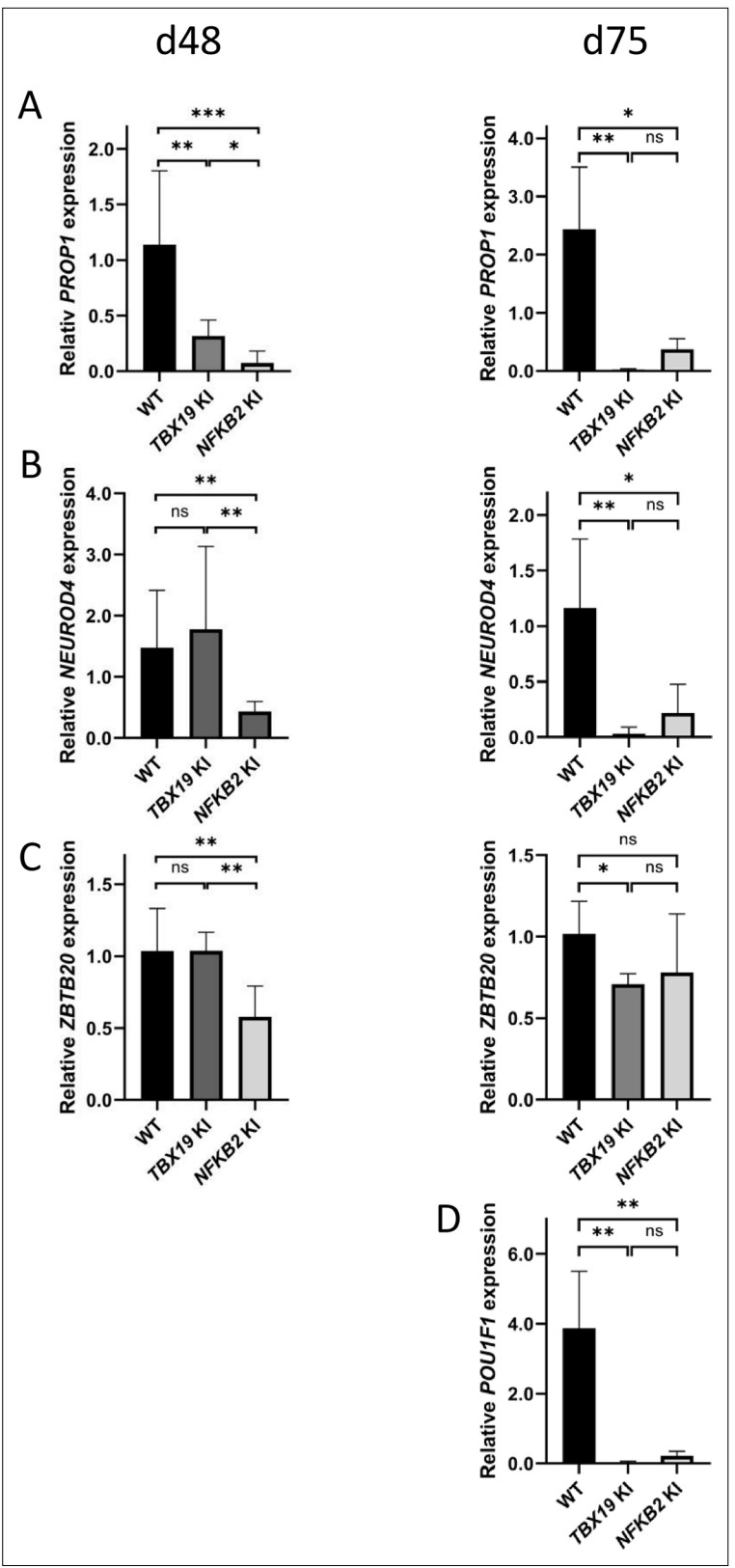

**Figure 12.** POU1F1-dependent lineages are affected in *TBX19* KI and nuclear factor kappa-B subunit 2 (*NFKB2)* KI organoids. (**A**) Strong downregulation of *PROP1* expression is observed at d48 in both mutants and persists at d75. (**B**) Expression of POU1F1-dependant lineage progenitors and mature somatotrophs marker *NEUROD4* is decreased at d48 inNFKB2 organoids, and in both models at d105. (**C**) *ZBTB20*, a marker for POU1F1-dependant

*Figure 12 continued on next page*

*Figure 12 continued*

lineage progenitors and mature lactotrophs, is less expressed in *NFKB2* KI organoids at d48, but at normal levels by d75. In *TBX19* KI mutants, its expression is normal at d48 but decreased by d75. (**D**) On d75, *POU1F1* expression is barely detectable in either model. Data show means ± SEM; d48: n=17, 12, and 11; d75: n=9, 7, and 6 for WT, *TBX19* KI and *NFKB2* KI organoids, respectively. Mann-Whitney t-test (unpaired, two-tailed, nonparametric). p<0.05 (*), p<0.01 (**), p<0.005 (***).

surprising that no pituitary deficiencies have been observed in *Nfkb2*[−/−] or *Nfkb2*[Lym1/Lym1]. This suggests that in humans, *NFKB2* plays an important role during pituitary development, therefore, constituting a major inter-species difference. Studying p52 binding regions across different species may help understand how this role appeared during evolution.

All patients with DAVID syndrome harbour heterozygous *NFKB2* mutations. Whether this reflects embryonic lethality of the mutation at the homozygous state or just the rarity of this genotype is still an unresolved question. However, 'super repressor' mutations have been shown to disturb both canonical and non-canonical signalling pathways even when heterozygous (*Wirasinha et al., 2021*; *Kuehn et al., 2017*). As our results show impaired POU1F1-dependent lineage development in addition to the severe corticotroph deficit, further experiments on organoids derived from *NFKB2*[WT/D865G] hiPSCs should tell if there are differences in the sensitivity of different lineages to the dominant-negative

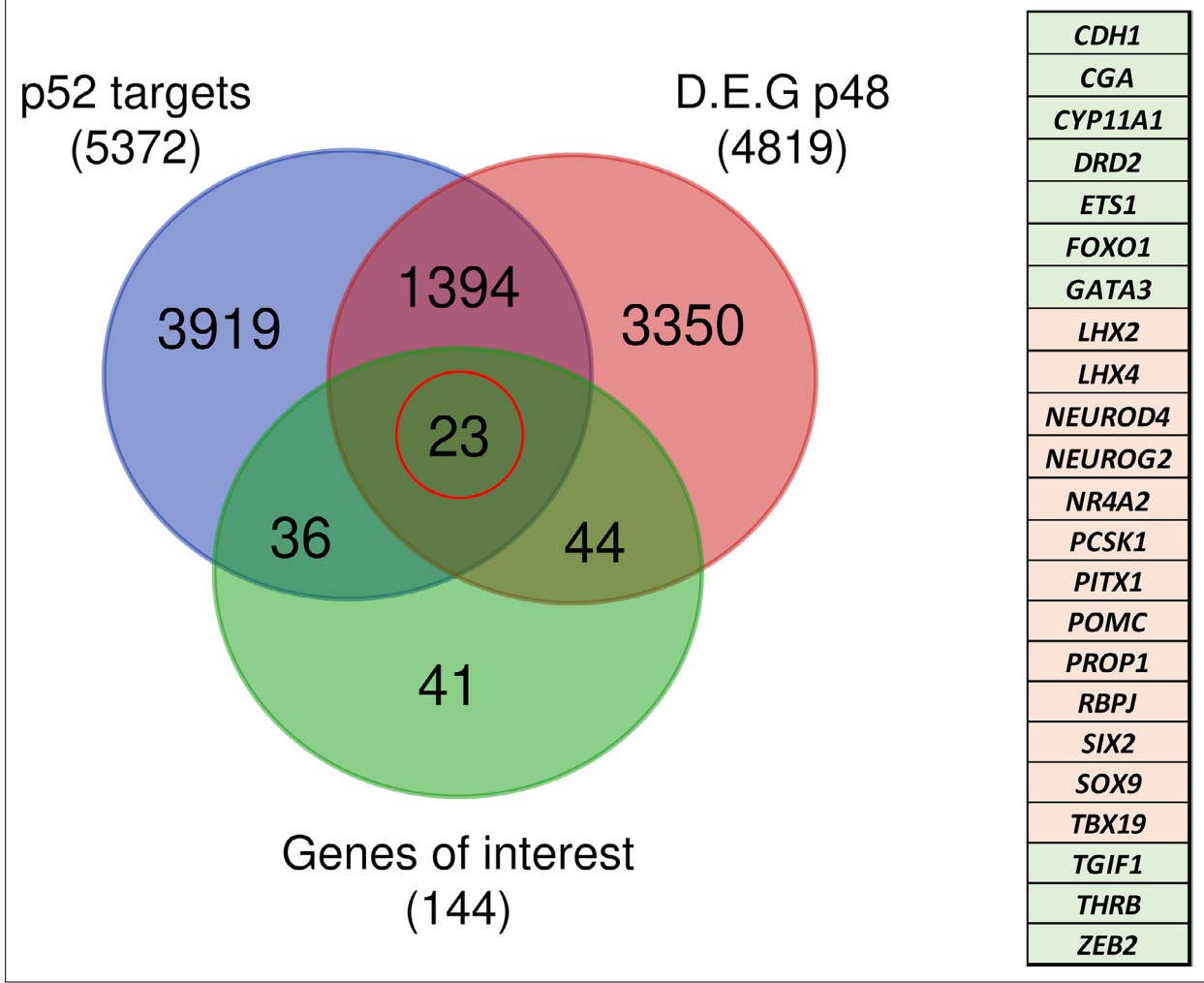

**Figure 13.** Identification of genes potentially under direct p52 regulation. Venn diagram showing the total number and intersections of genes identified as p52 targets, genes differentially expressed at d48 between wild-type (WT) and nuclear factor kappa-B subunit 2 (*NFKB2*) KI organoids, and a literature-curated list of genes particularly implicated in hypothalamus-pituitary development. The table on the right shows the identity of the 23 genes (circled in red), with color code indicating whether they are upregulated (green) or downregulated (pink) in *NFKB2* KI mutants at d48.

protein. If so, this could give us clues about why DAVID patients suffer from isolated ACTHD in their vast majority rather than CPHD.

Despite some technical limits, CRISPR/Cas9-based genome editing of hiPSC is a powerful approach to elucidate gene function in congenital pituitary deficiency patients, as was shown for the role of hypothalamic OTX2 regulation of pituitary progenitor cells in congenital pituitary hypoplasia (*Matsumoto et al., 2019*). While the prevalence of pathogenic variants of known genes in primary hypopituitarism is currently estimated at below 10% (*Jullien et al., 2021*), exome and whole-genome sequencing regularly identify new genes and variants, for which pathogenicity remains uncertain (*Dobin et al., 2013*). This model could thus become in the following years the gold standard to confirm the involvement of a given gene in pituitary development, or to classify new variants of unknown significance as likely benign or pathogenic. Finally, pituitary organoids derived from human embryonic stem cells have been transplanted subcutaneously into mice with hypopituitarism; grafted cells were able to secrete ACTH and respond to corticotropin-releasing hormone stimulation (*Sasaki et al., 2023*), paving the way for innovative substitutive treatments in patients with hypopituitarism (*Taga et al., 2023*). Conversely, a limitation of this model is the long duration of the differentiation period (approximately 3 months) and the fact that not all hiPSC clones lead to full differentiation of hypothalamic-pituitary organoids despite similar conditions of culture. For these reasons, we could not include confirmation of our results on an independent clone in the present paper.

In conclusion, we have designed and validated a model replicating human constitutive ACTHD using a combination of CRISPR/Cas9 and directed differentiation of hiPSC into 3D hypothalamic-pituitary organoids. Not only has this proof-of-concept reproduced the known prerequisite for the TBX19 transcription factor in human corticotroph differentiation, but it has also allowed a new classification of a *NFKB2* variant of previously unknown clinical significance as pathogenic in pituitary development.

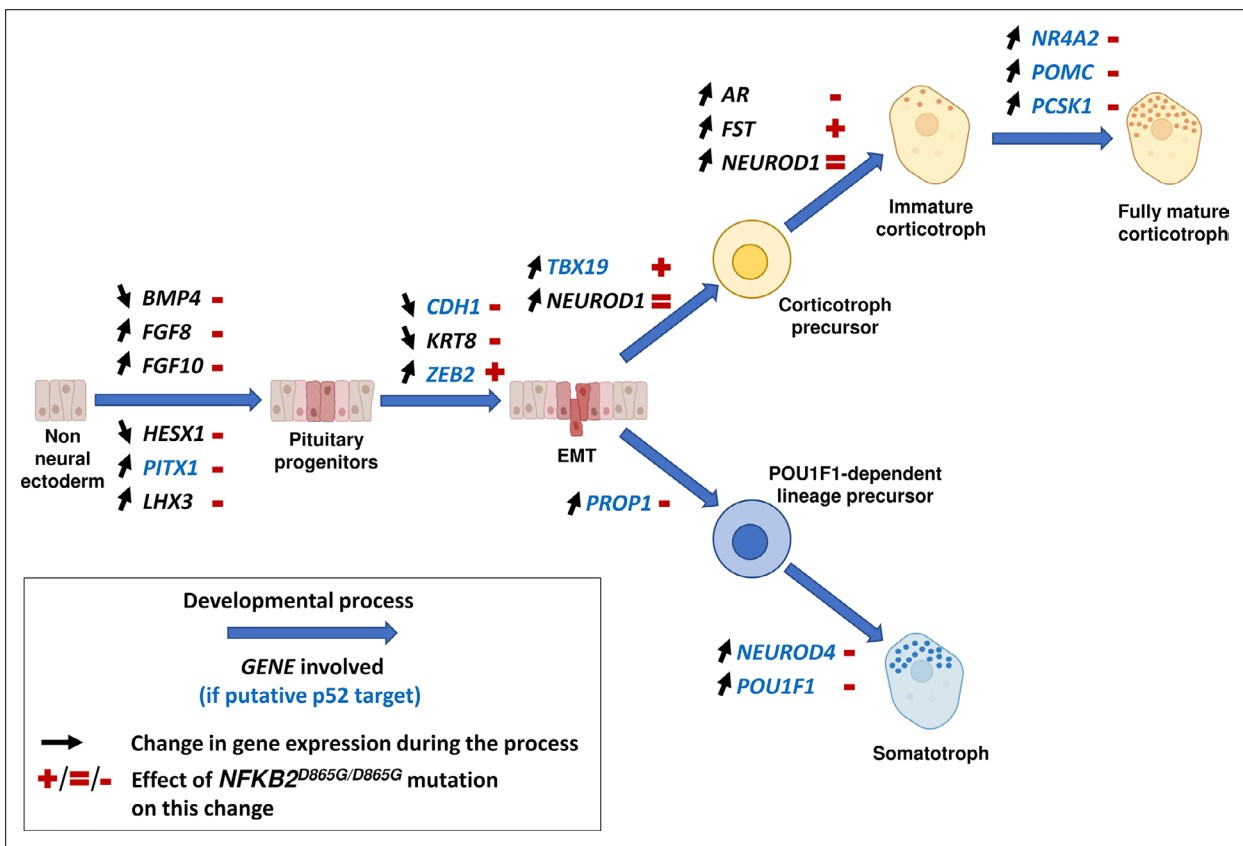

**Figure 14.** Changes in gene expression induced by nuclear factor kappa-B subunit 2 (*NFKB2*)$^{D865G/D865G}$ mutation during hypothalamic-pituitary organoid development. Diagram illustrating changes in gene expression (black arrows) associated with different stages of normal hypothalamic-pituitary organoids development and cell differentiation (large blue arrows). The *NFKB2*$^{D865G/D865G}$ mutation perturbs these changes (+/=/- signs in red), possibly through direct transcriptional control for genes indicated in blue.

From a clinical viewpoint, this organoid model could provide evidence for the pathogenic role of new candidate genes or variants investigated in patients where none other is available, especially in a rare disease affecting a poorly accessible organ like the pituitary gland. From a physiological viewpoint, modeling corticotroph deficiency using hiPSC allows us to find evidence for a functional role for *NFKB2* in human pituitary differentiation.

Future studies investigating the direct role of p52 on transcription and changes in chromatin landscape during pituitary development will be necessary to elucidate its precise mechanism of action, as well as the level of pathogenicity of the *NFKB2*$^{D865G}$ mutations in its heterozygous form.

# Materials and methods

## Key resources table

| Reagent type (species) or resource | Designation | Source or reference | Identifiers | Additional information |
|---|---|---|---|---|
| Cell line (*Homo sapiens*) | Human induced pluripotent stem cells | MaSC Platform, MMG UMR1251, INSERM-AMU | 10742 L | Reprogrammed from fibroblasts (Coriell, female, 82yo) |
| Other | Synthemax II-SC Substrate | Corning | #3335 | Cell culture matrix |
| Other | StemMACS hiPSC-Brew XF human medium | MACS Miltenyi Biotec | #130-104-368 | Cell culture medium |
| Other | Knockout Serum Replacement | Thermo Fisher Scientific | #10828028 | Cell culture reagent |
| Recombinant protein | Human BMP-4 | Peprotech | #AF-120-05ET | |
| Drug | Smoothened Agonist | Sigma-Aldrich/Merck | #SML 1314 | |
| Peptide | AlbumiNZ Bovine Albumin Low IgG | MP Biomedicals | #0219989790 | |
| Sequence-based reagent | ligonucleotides used for CRISPR/Cas9 experiments to target *TBX19*$^{K146R}$ | This paper | *TBX19* sgRNA | AAGCTGACCAACAAGCTCAA |
| Sequence-based reagent | ligonucleotides used for CRISPR/Cas9 experiments to target *TBX19*$^{K146R}$ | This paper | *TBX19* sgRNA oligo up | 5'-CACCGAAGCTGACCAACAAGCTCAA-3' |
| Sequence-based reagent | ligonucleotides used for CRISPR/Cas9 experiments to target *TBX19*$^{K146R}$ | This paper | *TBX19* sgRNA oligo down | 5'-AAACTTGAGCTTGTTGGTCAGCTTC-3' |
| Sequence-based reagent | ligonucleotides used for CRISPR/Cas9 experiments to target *TBX19*$^{K146R}$ | This paper | *TBX19* ssODN | TGGATGAAAGCTCCCATCTCCTTCAG CAAAGTGAGGCTGACCAACAAGTTaAA TGGAGGCGGGCAGGTACGAATGAGG CGGGCAGGCCTGGCCACCCGCT |
| Sequence-based reagent | primers used for PCR analyses, CAPS assay, and Sanger Sequencing | This paper | *TBX19* fwd | CCCCTGGACAAGGTGAGAGTT |
| Sequence-based reagent | primers used for PCR analyses, CAPS assay, and Sanger Sequencing | This paper | *TBX19* rev | GACTCCCGGGAATAATTGGCTTC |
| Sequence-based reagent | ligonucleotides used for CRISPR/Cas9 experiments to target *NFKB2*$^{D865G}$ | This paper | *NFKB2* sgRNA | GTGAAGGAAGACAGTGCGTA |
| Sequence-based reagent | ligonucleotides used for CRISPR/Cas9 experiments to target *NFKB2*$^{D865G}$ | This paper | *NFKB2* sgRNA oligo up | 5'-CACCGTGAAGGAAGACAGTGCGTA-3' |
| Sequence-based reagent | ligonucleotides used for CRISPR/Cas9 experiments to target *NFKB2*$^{D865G}$ | This paper | *NFKB2* sgRNA oligo down | 5'-AAACTACGCACTGTCTTCCTTCAC-3' |
| Sequence-based reagent | ligonucleotides used for CRISPR/Cas9 experiments to target *NFKB2*$^{D865G}$ | This paper | *NFKB2* ssODN | TCCCATTCCTGTCCCCATTTACCCCC AGCAGAGGTGAAGGAAGGCAGTGCC TACGGGAGCCAGTCAGTGGAGCAGG AGGCAGAGAAGCTGGGCCCACCCC |
| Sequence-based reagent | primers used for PCR analyses, CAPS assay, and Sanger Sequencing | This paper | *NFKB2* fwd | CCCTAACCATGACTCAGACCTCA |
| Sequence-based reagent | primers used for PCR analyses, CAPS assay, and Sanger Sequencing | This paper | *NFKB2* rev | CCTCCCCTTCCCATGAGAATCC |
| Antibody | Anti-human NFKB2 (p100/p52) Rabbit polyclonal | Sigma, | HPA008422 RRID:AB_1854434 | (1/500) |
| Antibody | Anti-human LHX3 Mouse monoclonal | DSHB | 67.4E12 RRID:AB_2135805 | (1/500) |
| Antibody | Anti-human TBX19 Rabbit polyclonal | Sigma, | HPA072686 RRID:AB_2732209 | (1/300) |
| Antibody | Anti-human ACTH Guinea pig polyclonal | NIDDK | | (1/2000) |

*Continued on next page*

*Continued*

| Reagent type (species) or resource | Designation | Source or reference | Identifiers | Additional information |
|---|---|---|---|---|
| Antibody | Anti-human E-human cadherin Rat monoclonal | Millipore, | MABT26 RRID:AB_10807576 | (1/500) |
| Antibody | Anti-human NKX2.1 Mouse monoclonal | Millipore, | MAB5460 RRID:AB_571072 | (1/500) |
| Antibody | Anti-human PITX1 Rabbit polyclonal | Sigma, | HPA008743 RRID:AB_1855413 | (1/500) |

## Culture and maintenance of hiPSC lines

The 10742 L healthy individual-derived hiPSC line was used in the study as the WT control (provided by the Cell Reprogramming and Differentiation Facility [MaSC], Marseille Medical Genetics, Marseille, France). Information about this hiPSC line is indicated in *Supplementary file 1a*. Two 'knock-in' mutant lines carrying $TBX19^{K146R/K146R}$ (thereafter also designated as $TBX19$ KI) and $NFKB2^{D865G/D865G}$ (thereafter also abridged as $NFKB2$ KI) were generated using CRISPR/Cas9 editing from the isogenic control line. All hiPSC lines were cultured on six-well plates (Corning, #3335, New York, USA) coated with Synthemax II-SC Substrate (working concentration at 0.025 mg/mL, Corning, New York, USA) and maintained undifferentiated in a chemically defined growth medium (StemMACS hiPSC-Brew XF human medium; MACS Miltenyi Biotec, Paris, France) (*Giobbe et al., 2015*). hiPSC lines were maintained in a humidified incubator under conditions of 37 °C, 5% $CO_2$, with a daily change of medium, and passaged when cells reached 60–80% confluency using enzyme-free ReLeSR according to manufacturer's recommendations (StemCell Technologies, Canada, #05872).

## CRISPR/Cas9 mediated genome editing of hiPSC

CRISPR/Cas9 gene editing was used to create the $TBX19^{K146R/K146R}$ and the $NFKB2^{D865G/D865G}$ mutations using the method as previously described (*Arnaud et al., 2022*).

CRISPR/Cas9 was used to generate two mutant hiPSC lines from 10742 L hiPSC:

1. The $TBX19^{K146R/K146R}$ line, harboring the c.437A>G, p.Lys146Arg $TBX19$ (NM_005149) pathogenic missense variant (*Couture et al., 2012*).
2. The $NFKB2^{D865G/D865G}$ line, harboring the c.2594A>G, p.Asp865Gly $NFKB2$ (NM_001322934) pathogenic missense variant.

## Preparation of the CRISPR/Cas9 sgRNA plasmid

For each targeted gene, sgRNA plasmid was prepared as previously described (*Ran et al., 2013*). Briefly, we designed a sgRNA sequence within 20 nucleotides from the target site selected with the open-source CRISPOR tool (http://crispor.tefor.net/) (*Concordet and Haeussler, 2018*).

The sgRNA sequence used to generate $TBX19^{K146R}$ was (5'>3'): AAGCTGACCAACAAGCTCAA (see also *Supplementary file 1b*).

The sgRNA sequence used to generate $NFKB2^{D865G}$ was (5'>3'): GTGAAGGAAGACAGTGCGTA (see also *Supplementary file 1c*).

We used the cAB03 (also known as pX459-pEF1alpha) vector as previously described (*Arnaud et al., 2022*). Briefly, pSpCas9 (BB)–2A-Puro (PX459) V2.0 (Addgene plasmid # 62988, Addgene, Teddington, UK) was modified by the EF1 alpha promoter (*Arnaud et al., 2022*). The chosen sgRNA was cloned into the cAB03 open plasmid vector using the method previously described by *Arnaud et al., 2022*. The final vector expressed the corresponding sgRNA under the control of the U6 promoter, as well as *Cas9* under the control of the EF-1alpha promoter, and a puromycin resistance gene.

## Single-stranded oligo DNA nucleotides (ssODN) design

We also designed donor sequences to generate $TBX19^{K146R}$ and $NFKB2^{D865G}$ missense mutations. A donor sequence is used for homology-directed repair, allowing the introduction ('knock-in,' KI) of single nucleotide variations, with low efficiency. To optimize the efficiency of KI editing, the donor sequences were designed as ssODN, carrying the mutation of interest, and also included a silent (or blocking) mutation that mutates the protospacer-adjacent motif (PAM) or disrupts the sgRNA binding

region to prevent re-cutting by Cas9 after successful editing (*Paquet et al., 2016*; *Edmondson et al., 2021*; *Arnaud et al., 2022*; *Kwart et al., 2017*). The ssODN contained 100 nucleotides with homology arms on each side of the target region. The ssODN sequence was as follows (see also *Supplementary file 1b* and *Supplementary file 1c*):

For *TBX19^{K146R}* (5'–3'): TGGATGAAAGCTCCCATCTCCTTCAGCAAAGTGAGGCTGACCAACAAG TTAAATGGAGGCGGGCAGGTACGAATGAGGCGGGCAGGCCTGGCCACCCGCT.

For *NFKB2^{D865G}* (5'–3'): TCCCATTCCTGTCCCCATTTACCCCCAGCAGAGGTGAAGGAAGGCAGTGCCTACGGGAGC CAGTCAGTGGAGCAGGAGGCAGAGAAGCTGGGCCCACCCC.

The ssODN were synthesized by Integrated DNA Technologies (Coralville, Iowa, US) at Ultramer quality.

## Electroporation

hiPSC were transfected with constructed sgRNA/Cas9 vectors by electroporation using the Neon Transfection System 100 µL kit (Invitrogen). A summary of the procedure is described in *Figure 1—figure supplement 1*.

## Clone isolation and screening

On day 3 post-transfection, we used a cleaved amplified polymorphic sequences (CAPS) assay to estimate the efficacy of CRISPR/Cas9 in bulk transfected hiPSC populations to choose the number of colonies to be picked (*Zischewski et al., 2017*). A restriction enzyme recognition site is inserted in the donor template. CAPS primers and restriction enzymes were determined with the help of AmplifX software (version 2.0.0b3). CAPS primers were listed in *Supplementary file 1d*. The CAPS products were separated and visualized by electrophoresis on a 2% agarose gel and analyzed for densitometry with ImageJ (*Arnaud et al., 2022*; *Kwart et al., 2017*).

After 10 d, the CAPS assay was also used to screen and identify the possible KI clones. KI clones were screened by CAPS assay and confirmed by Sanger sequencing (Genewiz, Leipzig, Germany). Analyses of Sanger traces were done by Sequencher DNA sequence analysis software (version 5.4.6, Gene Codes Corporation, Ann Arbor, MI USA, http://www.genecodes.com) to align sequences of KI clones with WT sequence to confirm the successful edition. Sanger sequencing primers are described *Supplementary file 1d*. KI clones were kept, amplified, and then cryopreserved.

## In vitro pituitary organoids differentiated from hiPSC

Pituitary organoids were produced at the same time, using the same protocol for WT and edited hiPSC lines. Two batches of differentiation with two hundred organoids for each hiPSC line were generated:

Batch 1: WT organoids versus *TBX19* KI organoids
Batch 2: WT organoids versus *NFKB2* KI organoids

The differentiation protocol was based on a protocol from Matsumoto et al. with some modification (*Matsumoto et al., 2019*; *Sasai et al., 2012*). hiPSCs were first dissociated into single cells using Accutase. Ten thousand cells per organoid were plated in 20 µL hanging drop under the lid of a 127.8×85.5 mm single-well plate (Greiner bio-one) in growth factor-free chemically defined medium (gfCDM) supplemented with 10% (vol/vol) Knockout Serum Replacement (KSR; Thermo Fisher Scientific) and 20 µM Y-27632. The lid was inverted over the bottom chamber (filled with 10 mL of sterile water to avoid the evaporation of droplets) and incubated at 37 °C/5% $CO_2$ for 2 d. Once aggregates formed, organoids were transferred to non-adhesive 100x20 mm culture dishes (Corning) containing 10 mL of gfCDM medium supplemented with 10% KSR without Y-27632 and incubated at 37 °C and 5% $CO_2$ until day 6. From days 6 to days 17, the medium was supplemented with 10% KSR, 5 nM recombinant human bone morphogenetic protein 4 (BMP4; Peprotech, # AF-120-05ET, USA) and 2 µM Smoothened agonist (SAG; Sigma-Aldrich/Merck, # SML 1314). Half of the medium was renewed every 2–3 d.

From day 18, BMP4 was withdrawn and half of the medium was renewed every 2–3 d. At this point, organoids were maintained under high-$O_2$ conditions (40%) and 5% $CO_2$ in an MCO-5M incubator (Panasonic). From day 30, gfCDM medium supplemented with 20% KSR was used, and all medium was renewed every 2–3 d until at least day 105. Induction into pituitary progenitor cells (defined as

LHX3+) and into pituitary hormone-producing cells was evaluated by qRT-PCR (on days 0, 6, 18, 27, 48, 75, 105) and immunofluorescence (on days 48 and 105) (see below).

## Validating *TBX19*$^{K146R}$ targeting and assessing off-target mutations by whole genome sequencing (WGS)

For *TBX19*$^{K146R}$, we used WGS to assess on-target and off-target mutations, because the edited site is far from the cut site (15 bases). DNA extracted from control and *TBX19* KI (5 organoids each) was submitted for WGS to an average depth of 15 X (Integraged Genomics platform). PCR-free libraries were prepared with the NEBNext Ultra II DNA Library Prep Kits (New England BioLabs) according to supplier recommendations. Specific double-strand gDNA quantification and a fragmentation (300 ng of input with high-molecular-weight gDNA) sonication method were used to obtain approximately 400 bp fragments. Finally, paired-end adaptor oligonucleotides (xGen TS-LT Adapter Duplexes from Integrated DNA Technologies) were ligated and re-paired. Tailed fragments were purified for direct sequencing without a PCR step. DNA PCR-free libraries were sequenced on paired-end 150 bp runs on the NovaSeq6000 (Illumina) apparatus. Image analysis and base calling were performed using the Illumina Real Time Analysis (RTA) Pipeline with default parameters. Sequence reads were mapped on the Human Genome Build (GRCh38) using the Burrows-Wheeler Aligner (BWA) (*Li and Durbin, 2009*). Variant calling was performed via the GATK Haplotype Caller GVCF tool (GATK 3.8.1, Broad Institute) (*Poplin et al., 2018*). Mutation enrichment was determined using Fisher's Exact Test. Variants were annotated with Ensembl's Variant Effect Predictor, (VEP) by Integragen Genomics (*McLaren et al., 2016*). NGS analyses confirmed the presence of on-target mutation and the absence of off-target mutation in the top four predicted off-target sites.

## Validating *NFKB2*$^{D865G}$ targeting, assessing off-target mutations, and differential expression analysis by bulk RNA sequencing (RNA-seq)

For *NFKB2*$^{D865G}$, we used bulk RNA-seq for analysis of differential RNA expression, and to assess on-target and off-target mutations, because the target site is close to the cleavage site (6 bases). A total of 500 ng of total RNA was extracted from 10 individual organoids (five *NFKB2* KI and five control organoids). RNA-seq libraries were generated using the KAPA mRNA HyperPrep kit (Roche). The quality and profile of the libraries were visualized on a Bioanalyzer using the High Sensitivity DNA assay (Agilent) and quantified on a Qubit using the dsDNA High Sensibility Kit (Thermo Fisher Scientific). Finally, we performed a 2×76 bp paired-end sequencing on a NextSeq500 (Illumina). After quality control using FastQC (Braham Institute), reads were aligned on the GRCh38 human reference genome using STAR (version 2.7.2b) (*Dobin et al., 2013*). BAM files were ordered and indexed using Samtools (*Danecek et al., 2021*). Read counts on genes were determined using StringTie (*Pertea et al., 2015*). We assessed the presence of desired *NFKB2* edits according to the CRISPR sgRNA design by direct visualization of the BAM files and the absence of off-target coding variations by the GATK Haplotype Caller GVCF tool (*Poplin et al., 2018*). Genes differentially expressed between conditions were identified using the R package DESeq2 (version 1.34.0) (*Love et al., 2014*) and assigned as such with an adjusted p-value <0.05.

## Total RNA isolation and quantitative real-time PCR (qRT-PCR) analysis for assessment of mRNA expression during pituitary organoid development

Organoids were harvested from the culture, placed on ice and then stored at –80 °C until RNA extraction. The total RNA was extracted from WT, *TBX19* KI, and *NFKB2* KI organoids using the NucleoSpin RNA Plus XS kit (Macherey-Nagel), and measured on a NanoDrop TM 1000 Spectrophotometer (Thermo Fisher Scientific). Organoid RNAs were extracted on days 0, 6, and 18 from 7 or 8 organoids, or on days 27, 48, 75, and 105 from single organoids (3–8 organoids per group for each time point). cDNA was synthesized from 200 ng total RNA using the mix of M-MLV reverse transcriptase (Invitrogen, UK, #28025013), dNTP 10 mM (Invitrogen, #10297018), RNAseOut Inhibitor (Invitrogen, #10777019) and random primers (Invitrogen, UK, #48190011). Quantitative PCR was performed using iTaq Universal SYBR Green Supermix (Bio-Rad Laboratories), on Quantstudio 5 Real-time PCR system (Thermo Fisher Scientific) and analysed using QuantStudio 5 Design & Analysis Software. The thermal cycling profiles were as follows: initial denaturation at 95 °C for 20 s, followed by 40 cycles of

denaturation at 95 °C (3 s), annealing at 60 °C (45 s), and extension at 72 °C (45 s). All samples were assayed in duplicate. The beta-actin (*ACTB*), glyceraldehyde-3-phosphate dehydrogenase (*GAPDH*), and beta-tubulin (*TUBB*) transcript expressions were used as three endogenous reference controls. Target gene expressions were normalized relative to the mean of the housekeeping genes using the delta Ct method. For gene expression kinetics, these values were normalized relative to the highest mean values of WT during 105 d using the comparative $2^{-\Delta\Delta Ct}$ method (*Livak and Schmittgen, 2001*). Primer sequences used in the experiments are shown in *Supplementary file 1g*.

## Immunofluorescence labeling experiments

At days 48 and 105, ten of each WT and mutant organoids were fixed using 4% buffered para-formaldehyde in standard phosphate-buffered saline (PBS) for an hour at RT before rinsing in PBS. After overnight incubation in 30% sucrose/PBS (w/v) at 4 °C, organoids were embedded in Surgipath medium (Leica) and snap frozen. Organoids were cryosectioned at 16 µm. After blocking with 0.01% Triton X100, 10% normal donkey serum, and 1% fish skin gelatin in PBS with 5 ng/mL 4',6-diamidino-2-phénylindole (DAPI) for an hour at RT, slides were incubated with primary antibody diluted in antibody solution (PBS, Triton as above, 1% normal donkey serum and 0.1% fish skin gelatin) overnight at 4 °C, washed and incubated in a 1/500 dilution of secondary antibody in PBS for 2 hr at RT. The primary antibodies and dilutions used in this study are summarized in *Supplementary file 1h*. Secondary antibodies were species-specific conjugates to Alexa Fluor 488, 555, or 647 (Thermo Fisher Scientific). Fluorescence images were obtained using LSM 800 confocal microscopy (Zeiss).

## 3D imaging of pituitary organoids

We used a whole-mount immunostaining and clearing protocol adapted from *Belle et al., 2017*. 4–8 organoids of each genotype (WT, *TBX19* KI, or *NFKB2* KI) were fixed by immersion in 4% buffered paraformaldehyde (PFA) in PBS overnight at 4 °C. All steps were carried out on a slowly rotating (70 rpm) agitator. After washing in PBS, organoids were either stored at 4 °C or blocked and permeabilized with 5 mL PBSGT for 2 d at RT (PBSGT: 0.2% fish skin gelatin, 0.5% Triton X100 in PBS, NaN₃ 0.01%). Organoids were incubated with primary antibodies in 3 mL PBSGT (at concentrations used on sections) over 5 d at 37 °C to increase penetration. Organoids were then washed 6 times in an excess of PBSGT at RT, before incubation with secondary antibodies in 3 ml PBSGT at RT overnight. The wash step was repeated. Organoids were then embedded in 1% low-melting temperature agarose in 1 X TAE after it had cooled to about 45 °C.

As adapted from the iDISCO+ protocol (*Renier et al., 2016*), organoids in agarose blocks were progressively dehydrated from 20%, 40%, 60%, and 80% to 100% methanol for 1 hr at each graded step. After overnight incubation in two parts dichloromethane (DCM, Sigma-Aldrich) to one part 100% methanol, organoids were immersed in 100% DCM for 30 min before changing to 100% dibenzyl ether (DBE; Sigma-Aldrich) for transparization over 2 hr. Samples were maintained in DBE at RT and protected from light in an amber glass vial. Light-sheet fluorescence microscopy (Miltenyi Biotec Ultra-Microscope Blaze) was used to acquire z-stacks of optical sections of pituitary organoids at 4 µm intervals, in Ethyl cinnamate (Sigma-Aldrich). After 3D reconstruction with Imaris software (version 9.6, Bitplane), the number of corticotrophs in each organoid was defined by counting individualized cell-sized (7 µm) objects positive for ACTH immunoreactivity with the Spots automatic classifier after setting parameters in a region of interest containing positively labeled WT cells, then applied to all organoids. Using the Surfaces tool of the Imaris software, the volume of organoids was determined based on thresholds corresponding to their respective fluorescence intensities.

## Statistical analysis

Statistical analyses were performed and visualized with GraphPad Prism 9.5.0 software (GraphPad Software, Inc). Data are expressed as means ± SEM. Comparisons between the two groups (mutant versus WT) were performed by unpaired two-tailed t-tests (non-parametric Mann-Whitney test). n refers to the number of samples for each experiment outlined in the figure legends. p-values of <0.05 (*), <0.01 (**), and <0.001 (***) were considered statistically significant differences.

## Acknowledgements

This study was supported by the ADEREM (Association pour le Développement de la Recherche Médicale au CHU de Marseille), the AFM-Telethon MoThARD grant to TB and by Pfizer (Grant ID#88979183). TTM received funding from France Excellence Scholarship of the French Embassy in Vietnam, from the French Society of Pediatric Endocrinology and Diabetology (SFEDP), and from the Marseille Rare Disease (MarMaRa) Institute of Aix-Marseille University. The authors wish to thank Frederique Magdinier, Natacha Broucqsault, and Claire El Yazidi from the Marseille Medical Genetics (MMG) cell reprogramming & differentiation facility (MaSC); Sébastien Courrier, from the MMG microscopy platform; Valerie Delague, Catherine Aubert, Christel Castro and Camille Humbert from MMG's Genomics and Bioinformatics (GBiM) platform; Carole Siret, and Mathieu Fallet from the Centre d'Immunologie de Marseille-Luminy (CIML); Ivo Vanzetta and Alberto Lombardini from the Institut de Neurosciences de la Timone (INT) microscopy platform, and Laurie Arnaud and Emmanuel Nivet from the Institut de Neurophysiopathologie (INP), all in Marseille, for their expert assistance, advice or reagents.

## Additional information

### Funding

| Funder | Grant reference number | Author |
|---|---|---|
| Pfizer France | ID#88979183 | Thierry Brue |
| AFMTéléthon | MOTHARD Project N° 23408 | Thierry Brue |

The funders had no role in study design, data collection and interpretation, or the decision to submit the work for publication.

### Author contributions

Thi Thom Mac, Conceptualization, Data curation, Formal analysis, Investigation, Methodology, Writing – original draft; Teddy Fauquier, Conceptualization, Data curation, Formal analysis, Supervision, Validation, Investigation, Methodology, Writing - review and editing; Nicolas Jullien, Conceptualization, Data curation, Formal analysis, Supervision, Validation, Methodology, Writing – original draft; Pauline Romanet, Heather Etchevers, Validation; Anne Barlier, Supervision, Validation; Frederic Castinetti, Conceptualization, Supervision, Validation; Thierry Brue, Conceptualization, Formal analysis, Supervision, Funding acquisition, Validation, Project administration, Writing - review and editing

### Author ORCIDs

Thi Thom Mac  https://orcid.org/0000-0002-3746-2669
Teddy Fauquier  https://orcid.org/0000-0003-1132-9402
Heather Etchevers  https://orcid.org/0000-0003-0201-3799
Thierry Brue  https://orcid.org/0000-0001-8482-6691

Reviewer #1 (Public review): https://doi.org/10.7554/eLife.90875.3.sa1
Reviewer #2 (Public review): https://doi.org/10.7554/eLife.90875.3.sa2
Reviewer #3 (Public review): https://doi.org/10.7554/eLife.90875.3.sa3
Author response https://doi.org/10.7554/eLife.90875.3.sa4

## Additional files

### Supplementary files

• Supplementary file 1. Detailed lists of cell lines, oligonucleotides, PCR settings and antibodies used in this article. (a) hiPSC control line used in the study. (b) List of oligonucleotides used for CRISPR/Cas9 experiments to target $TBX19^{K146R}$. (c) List of oligonucleotides used for CRISPR/Cas9 experiments to target $NFKB2^{D865G}$. (d) List of primers used for PCR analyses, CAPS assay, and Sanger Sequencing. (e) PCR settings for $TBX19$. (f) PCR settings for $NFKB2$. (g) List of primers used for qRT-PCR. (h) List of primary antibodies used for immunostainings.

• MDAR checklist

## Data availability

RNA-seq results are included as a supporting file (*Figure 9—source data 1*).

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
