## [Editor Report · eLife Assessment]

This **important** study examines the effects of NFKB2 mutations on pituitary gland development through hypothalamic-pituitary organoids. The evidence supporting the main conclusions is **solid**, although analysis of additional clones to exclude inter-clone variability would strengthen the conclusions. This is a revised study, but insight into the mechanism of action of NFKB2 during pituitary development is incomplete. This work will be of interest to endocrinologists and biologists working on pituitary gland development and disease.

---

## [Referee Report · Reviewer #1 (Public review)]

Summary:

NFKB mutations are thought to be one of the causes of pituitary dysfunction, but until now they could not be reproduced in mice and their pathomechanism was unknown. The authors used the differentiation of hypothalamic-pituitary organoids from human pluripotent stem cells to recapitulate the disease in human iPS cells carrying the NFKB mutation.

Strengths:

The authors achieved their primary goal of recapitulating the disease in human cells. In particular, the differentiation of the pituitary gland is closely linked to the adjacent hypothalamus in embryology, and the authors have again shown that this method is useful when the hypothalamus is suspected to be involved in pituitary abnormalities caused by genetic mutations.

Weaknesses:

On the other hand, the pathomechanism is still not fully understood. This study provides some clues to the pathomechanism, but further analysis of NFKB expression and experiments investigating the relevant factors in more detail may help to clarify it further.

As for the revised manuscript, it is still insufficient for understanding the role of NFKB2 in pituitary development although their additional experiments have improved the manuscript. The strength of the hypothalamus-pituitary organoid lies in its ability to recapitulate the differentiation process including not only the pituitary cells but also neighbouring non-pituitary cells, such as hypothalamic cells in vitro. It is necessary to determine "at which stages" and "in which localizations" NFKB2 expression is critical for pituitary development.

---

## [Referee Report · Reviewer #2 (Public review)]

Summary:

DAVID syndrome is a rare autosomal dominant disorder characterized by variable immune dysfunction and variable ACTH deficiency. Nine different families have been reported, and all have heterozygous mutations in NFKB2. The mechanism of NFKB2 action in the immune systems has been well-studied, but nothing is known about its role in pituitary gland.

The DAVID mutations cluster in the C-terminus of the NFKB2 and interfere with cleavage and nuclear translocation. The mutations are likely dominant negative, by affecting dimer function. ACTH deficiency can be life-threatening in neonates and adults, thus, understanding the mechanism of NFKB2 action in pituitary development and/or function is important.

The authors use CRISPR/Cas gene editing of human iPSC derived pituitary-hypothalamic organoids to assess the function of NFKB2 and TBX19 in pituitary development. Mutations in TBX19 are the most common, known cause of pituitary ACTH deficiency, and the mechanism of action has been studied in mice, which phenocopy the human condition. Thus, the TBX19 organoids can serve as a positive control. The Nfkb2 mouse model has a p.Y868* mutation that impairs cleavage of NFKB2 p100, and the immune phenotype mimics the patients with DAVID mutations, but no pituitary phenotype was evident. Thus, a human organoid model might be the only approach suitable to discover the etiology of the pituitary phenotype.

Overall, the authors have selected an important problem, and the results suggest that the pituitary insufficiency in DAVID syndrome is caused by a developmental defect rather than an autoimmune hypophysitis condition. The use of gene editing in human iPSC derived hypothalamic-pituitary organoids is significant, as there is only one example of this previously, namely studies on OTX2. Only a few laboratories have demonstrated the ability to differentiate iPSC or ES cells to these organoids, and the authors have improved the efficiency of differentiation, which is also significant.

The strength of the evidence is excellent. The authors have thoroughly analyzed the genetically engineered organoids compared to isogenic controls. They have validated their findings with analysis of RNA and proteins. They have studied the time course of differentiation in the organoids and have a robust experimental design involving many replicates. Analysis of additional clones could strengthen the evidence.

Strengths:

The authors make mutations in TBX19 and NFKB2 that exist in affected patients. The TBX19 p.K146R mutation is recessive and causes isolated ACTH deficiency. Mutations in this gene account for 2/3 of isolated ACTH deficiency cases. The NFKB2 p.D865G mutation is heterozygous in a patient with recurrent infections and isolated ACTH deficiency. NFKB2 mutations are a rare cause of ACTH deficiency, and they can be associated with loss of other pituitary hormones in some cases. However, all reported cases are heterozygous.

The developmental studies of organoid differentiation are rigorous in that 200 organoids were generated for each hiPSC line, and 3-10 organoids were analyzed for each time point and genotype. Differentiation analysis relied on both RNA transcript measurements and immunohistochemistry of cleared organoids using light sheet microscopy. Multiple time points were examined, including seven times for gene expression at the RNA level and two times in the later stages of differentiation for IHC.

TBX19 deficient organoids exhibit reduced levels of PITX1, LHX3, and POMC (ACTH precursor) expression at the RNA and IHC level, and there are fewer corticotropes in the organoids, as ascertained by POMC IHC.

The NFKB2 deficient organoids have normal expression of the early pituitary transcription factor HESX1, but reduced expression of PITX2, LHX3 and POMC. Because there is no immune component in the organoid, this shows that NFKB2 mutations can affect corticotrope differentiation to produce POMC. RNA sequencing analysis of the organoids reveals potential downstream targets of NFKB2 action, including a potential effect on epithelial to mesenchymal like transition and selected pituitary and hypothalamic transcription factors and signaling pathways.

It is important to note that all NFKB2 patients are heterozygous for what appear to be dominant negative mutations that affect protein cleavage and nuclear localization of processed protein as homo or heterodimers. The organoids are homozygous for this mutation.

Weakness:

There could be variation between individual iPSC lines that is unrelated to the genetically engineered change. The work would be strengthened by analysis of independently engineered clones or by correcting the engineered clone to wild type and demonstrating that the phenotypic effects are reversed. The authors do check for off target effects of the guide RNA at predicted sites using WGS.

---

## [Referee Report · Reviewer #3 (Public review)]

Summary:

This manuscript by Mac et al addresses the causes of pituitary dysfunction in patients with DAVID syndrome which is caused by mutations in the NFKB2 gene and leads to ACTH deficiency. The authors seek to determine whether the mutation directly leads to altered pituitary development, as opposed to an autoimmune defect, by using mutating human iPSCs and then establishing organoids that differentiate into pituitary tissue. They first seek to validate the system using a well-characterised mutation of the transcription factor TBX19, which also results in ACTH deficiency in patients. Then they characterise altered pituitary cell differentiation in mutant NFKB2 organoids and show that these lack corticotrophs, which would lead to ACTH deficiency. Importantly, the findings here suggest the effects of mutant NFKB2 on pituitary organoid differentiation are direct and not a result of altered noncanonical NF-κB signalling, which has been shown to be a mechanism leading to immunodeficiency in DAVID patients.

Strengths:

The conclusion of the paper that ACTH deficiency in DAVID syndrome is independent of an autoimmune input is strong.

Weaknesses:

(1) The authors correctly emphasise the importance of establishing the validity of an iPSC-based model in being able to recapitulate in vivo dysfunctional pituitary development through characterisation of a TBX19 knock-in mutation. Whilst this leads to the expected failure of functional corticotroph differentiation, other aspects of the normal pituitary differentiation pathway upstream of cortocotroph commitment seem to have been affected in surprising ways. In particular, the loss of LHX3 and PITX1 in TBX19 mutant organoids compared with wild type requires explanation, especially as the mutant protein would only be expected to be expressed in a small proportion of anterior pituitary lineage cells. This may identify a difference between human and mouse pituitary development and emphasises the importance of further establishing the developmental programme in human pituitary.

(2) It is notable that the manipulation of iPSC cells used to generate mutants through CRISPR/Cas9 editing is not applied to the control iPSC line. It is possible that these manipulations, including electroporation and puromycin selection may lead to changes to the iPSC cells that is independent of the mutations introduced and this may change the phenotype of the cells. The authors have established that there are no off-target mutations through whole genome sequencing but the iPSC manipulation could have led to changes through epigenetic mechanisms or through non-genomic alterations of developmental potential. A better control in all experiments would have been an iPSC line with a benign knock-in (such as GFP into the ROSA26 locus) or use of a selected line where editing failed. The authors also ackowledge that use of a single clone is not ideal in these studies and characterisation of multiple clones would strengthen the conclusions of the study.

---

## [Author Response]

The following is the authors’ response to the original reviews.

**eLife Assessment**
This valuable study examines the effects of NFKB2 mutations on pituitary gland development through hypothalamic-pituitary organoids. The evidence supporting the main conclusions is solid, although analysis of additional clones to exclude inter-clone variability would strengthen the conclusions. Insight into the mechanism of action of NFKB2 during pituitary development is incomplete. This work will be of interest to endocrinologists and biologists working on pituitary gland development and disease.

We agree with these considerations and the summary and thank the Editors for their assessment. Although we indeed share the idea that reproduction of the experiments on a second clone would be a useful confirmatory step, we have not been able to reach this goal within a reasonable time frame for the reason mentioned above (unavailability of the main research engineer knowledgeable in the challenging methods involved for organoids differentiation) and due to the long turnaround time of this kind of experiments (3 months for the whole differentiation starting form iPSC). We therefore decided to publish on a single clone while we are still aiming at reproducing our results on at least a second one and will hopefully be able to provide these additional data in a subsequent revised version. We now acknowledge this limitation in the final part of the Discussion.

Revised text: “Conversely, a limitation of this model is the long duration of the differentiation period (approximately 3 months) and the fact that not all hiPSC clones lead to full differentiation of hypothalamo-pituitary organoids despite similar conditions of culture. For these reasons, we could not include confirmation of our results on an independent clone in the present paper.”

**Public Reviews:**

**Reviewer #1 (Public Review):**
Summary:NFKB mutations are thought to be one of the causes of pituitary dysfunction, but until now they could not be reproduced in mice and their pathomechanism was unknown. The authors used the differentiation of hypothalamic-pituitary organoids from human pluripotent stem cells to recapitulate the disease in human iPS cells carrying the NFKB mutation.Strengths:The authors achieved their primary goal of recapitulating the disease in human cells. In particular, the differentiation of the pituitary gland is closely linked to the adjacent hypothalamus in embryology, and the authors have again shown that this method is useful when the hypothalamus is suspected to be involved in pituitary abnormalities caused by genetic mutations.Weaknesses:On the other hand, the pathomechanism is still not fully understood. This study provides some clues to the pathomechanism, but further analysis of NFKB expression and experiments investigating the relevant factors in more detail may help to clarify it further.

We thank this reviewer for acknowledging that we've reached our primary objective, in particular the fact that the HPO (hypothalamo-pituitary organoid) model allows recapitulation of the disease in human cells, including hypothalamic-pituitary interactions. Regarding the pathophysiological mechanism of the disease, we must admit that it remains incompletely understood. However, we have analysed more samples by RT-qPCR and further analysed RNASeq data from *NFKB2* KI organoids, which provided with more insights into the different levels where NFKB2 may play a role. We have now provided several additional figures derived from these analyses, including a synthetic figure to summarize the most relevant observed effects (Fig. 14).

**Reviewer #2 (Public Review):**

We also thank this reviewer for the detailed analysis of our manuscript, for the valuable comments, suggestions and questions that are addressed point-by point below.

Summary:DAVID syndrome is a rare autosomal dominant disorder characterized by variable immune dysfunction and variable ACTH deficiency. Nine different families have been reported, and all have heterozygous mutations in NFKB2. The mechanism of NFKB2 action in the immune systems has been well-studied, but nothing is known about its role in the pituitary gland.The DAVID mutations cluster in the C-terminus of the NFKB2 and interfere with cleavage and nuclear translocation. The mutations are likely dominant negative, by affecting dimer function. ACTH deficiency can be life-threatening in neonates and adults, thus, understanding the mechanism of NFKB2 action in pituitary development and/or function is important.The authors use CRISPR/Cas gene editing of human iPSC-derived pituitary-hypothalamic organoids to assess the function of NFKB2 and TBX19 in pituitary development. Mutations in TBX19 are the most common, known cause of pituitary ACTH deficiency, and the mechanism of action has been studied in mice, which phenocopy the human condition. Thus, the TBX19 organoids can serve as a positive control. The Nfkb2<Lym1/Lym1> mouse model has a p.Y868* mutation that impairs cleavage of NFKB2 p100, and the immune phenotype mimics the patients with DAVID mutations, but no pituitary phenotype was evident. Thus, a human organoid model might be the only approach suitable to discover the etiology of the pituitary phenotype.Overall, the authors have selected an important problem, and the results suggest that the pituitary insufficiency in DAVID syndrome is caused by a developmental defect rather than an autoimmune hypophysitis condition. The use of gene editing in human iPSC-derived hypothalamic-pituitary organoids is significant, as there is only one example of this previously, namely studies on OTX2. Only a few laboratories have demonstrated the ability to differentiate iPSC or ES cells to these organoids, and the authors have improved the efficiency of differentiation, which is also significant.The strength of the evidence is excellent. However, the two ACTH-deficient organoid models use a single genetically engineered clone, and the potential for variability amongst clones makes the conclusions less compelling. Since the authors obtained two independent clones for NFKB2 it is not clear why only one clone was studied.

We experienced difficulties obtaining an hiPSC population devoid of spontaneous differentiation while purifying this second clone, and did not want to delay the start of the experiments. This clone will be analysed in a follow-up study.

Finally, the effect of TBX19 on early pituitary fate markers is somewhat surprising given the phenotype of the knockout mice and patients with mutations. Thus, the use of a single clone for that study is also worrisome.

We agree that the effect of the TBX19 mutant on early pituitary progenitor development is rather puzzling. In our model, TBX19 is expressed throughout the whole experiment, although it is at very low levels in undifferentiated hiPSCs compared to peak expression (over 50-fold difference).

During the CRISPR-Cas9 gene edition, we obtained a clone with a homozygous one base insertion at the cutting site, leading to a frameshift and a premature stop codon 48 bases downstream. This would result in an expected protein of 163 amino acids instead of 488, but with potentially still functional DNA-binding ability. This mutation had a similar effect on *LHX3* and *PITX1* as the *TBX19* KI mutation, although it was even more severe. Our most likely explanation is that the two TBX19 mutants we generated have dominant negative effects. Contrary to mouse, little is known about *TBX19* expression in early human pituitary development, but scRNA-seq data on human embryonic pituitaries (Zhang et al.) show low expression in undifferentiated pituitary progenitors between 7 and 9 weeks of gestation. Therefore, early expression of these dominant negative proteins could perturb differentiation in the organoids. Future development of hiPSCs lines with total absence of TBX19 should help clarify these questions.

Strengths:The authors make mutations in TBX19 and NFKB2 that exist in affected patients. The TBX19 p.K146R mutation is recessive and causes isolated ACTH deficiency. Mutations in this gene account for 2/3 of isolated ACTH deficiency cases. The NFKB2 p.D865G mutation is heterozygous in a patient with recurrent infections and isolated ACTH deficiency. NFKB2 mutations are a rare cause of ACTH deficiency, and they can be associated with the loss of other pituitary hormones in some cases. However, all reported cases are heterozygous.The developmental studies of organoid differentiation seem rigorous in that 200 organoids were generated for each hiPSC line, and 3-10 organoids were analyzed for each time point and genotype. Differentiation analysis relied on both RNA transcript measurements and immunohistochemistry of cleared organoids using light sheet microscopy. Multiple time points were examined, including seven times for gene expression at the RNA level and two times in the later stages of differentiation for IHC.TBX19 deficient organoids exhibit reduced levels of PITX1, LHX3, and POMC (ACTH precursor) expression at the RNA and IHC level, and there are fewer corticotropes in the organoids, as ascertained by POMC IHC.The NFKB2 deficient organoids have a normal expression of the early pituitary transcription factor HESX1, but reduced expression of PITX2, LHX3, and POMC. Because there is no immune component in the organoid, this shows that NFKB2 mutations can affect corticotrope differentiation to produce POMC. RNA sequencing analysis of the organoids reveals potential downstream targets of NFKB2 action, including a potential effect on epithelial-to-mesenchymal-like transition and selected pituitary and hypothalamic transcription factors and signaling pathways.Weaknesses:There could be variation between individual iPSC lines that is unrelated to the genetically engineered change. While the authors check for off-target effects of the guide RNA at predicted sites using WGS, a better control would be to have independently engineered clones or to correct the engineered clone to wild type and show that the phenotypic effects are reversed.All NFKB2 patients are heterozygous for what appear to be dominant negative mutations that affect protein cleavage and nuclear localization of processed protein as homo or heterdimers. The organoids are homozygous for this mutation. Supplemental Figure 4 indicates that one heterozygous clone and two homozygous mutant clones were obtained. Analysis of these additional clones would give more strength to the conclusions, showing reproducibility and the effect of mutant gene dosage.

The main goal of this work was to evaluate if and how *NFKB2D865G* mutation affects hypothalamic-pituitary organoids development, in order to determine if these organoids would constitute a valuable model to study DAVID syndrome.

We thank this reviewer for noting that we identified an important question and have used appropriate novel and not widely used methods to address it, including CRISPR/Cas9 genome editing of iPSCs and disease modelling in iPSC-derived HPOs that had not previously been reported by a team other than the one that initially described it, allowing to confirm our working hypothesis that DAVID syndrome is caused by a developmental defect rather than an autoimmune hypophysitis condition. We also agree that analysing more clones, generated from same or different hiPSC lines, carrying homozygous or heterozygous mutations, and corrected mutations will be necessary in the future.

**Reviewer #3 (Public Review):**

We also thank this reviewer for the detailed analysis of our manuscript, for the valuable comments, suggestions and questions that are addressed point-by point below.

Summary:This manuscript by Mac et al addresses the causes of pituitary dysfunction in patients with DAVID syndrome which is caused by mutations in the NFKB2 gene and leads to ACTH deficiency. The authors seek to determine whether the mutation directly leads to altered pituitary development, as opposed to an autoimmune defect, by using mutating human iPSCs and then establishing organoids that differentiate into pituitary tissue. They first seek to validate the system using a well-characterised mutation of the transcription factor TBX19, which also results in ACTH deficiency in patients. Then they characterise altered pituitary cell differentiation in mutant NFKB2 organoids and show that these lack corticotrophs, which would lead to ACTH deficiency.Strengths:The conclusion of the paper that ACTH deficiency in DAVID syndrome is independent of an autoimmune input is strong.Weaknesses:(1) The authors correctly emphasise the importance of establishing the validity of an iPSC-based model in being able to recapitulate in vivo dysfunctional pituitary development through characterisation of a TBX19 knock-in mutation. Whilst this leads to the expected failure of functional corticotroph differentiation, other aspects of the normal pituitary differentiation pathway upstream of corticotroph commitment seem to have been affected in surprising ways. In particular, the loss of LHX3 and PITX1 in TBX19 mutant organoids compared with wild type requires explanation, especially as the mutant protein would only be expected to be expressed in a small proportion of anterior pituitary lineage cells.If the developmental expression profile of key transcription factors in mutant organoids does not recapitulate that which occurs in vivo, any interpretation of the relevance of expression differences in the NFKB2 organoids to the mechanism(s) leading to corticotroph function in vivo has to be questionable.

See response to Reviewer #2

It is notable that the manipulation of iPSC cells used to generate mutants through CRISPR/Cas9 editing is not applied to the control iPSC line. It is possible that these manipulations lead to changes to the iPSC cells that are independent of the mutations introduced and this may change the phenotype of the cells. A better control would have been an iPSC line with a benign knock-in (such as GFP into the ROSA26 locus).

We agree that the issue of off-target mutations should be addressed. However, we performed whole genome sequencing on *TBX19* KI and did not observe any pathogenic variants other than the intended edition. We also checked that clones isolated during the screening procedure but that returned negative for editing still had the ability to generate pituitary cells. However, we made the choice to use the isogenic original hiPSC line as it could be compared to both TBX19 *KI* and NFKB2 *KI* simultaneously, therefore reducing workload and cost of the experiments. Any other knock-in mutation, such as GFP into the ROSA26 locus would imply the same risk of off-target mutations, but presumably at other sites in the genome.

(2) In the results section of the manuscript the authors acknowledge that hypothalamic tissue in the NFKB2 mutant organoid may be having an effect on the development of pituitary tissue. However, in the discussion the emphasis is entirely on pituitary autonomous mechanisms such as pituitary HESX1 expression or POMC gene regulation; in the conclusion of the abstract, a direct role for NFKB2 in pituitary differentiation is described. Whilst the data here may suggest a non-immune mediated alteration in pituitary function in DAVID syndrome, if this is due to alteration of the developing hypothalamus then this is not direct. A fuller discussion of the potential hypothalamic contribution and/or further characterisation of this aspect is warranted.

We agree with this reviewer that contributions of both hypothalamic and pituitary developing tissues should be taken into account. We performed more experiments and analysed the effect of both mutations on hypothalamic growth factors expression. These results are displayed in new figure 10. The role of the hypothalamus is now clearly mentioned and highlighted in the Discussion.

(3) qRT-PCR data presented in Figure 6A shows negligible alteration of HESX1 expression at all time points in NFKB2 mutant organoids. This is not consistent with the 2-fold increase in HESX1 expression described in day 48 organoids found by bulk RNA sequencing.How do the authors reconcile these results and why is one result focused on in the discussion where a potential mechanism for a blockade of normal pituitary cell differentiation is suggested? Further confirmation of HESX1 expression is required.

In the previous version on the manuscript, the *HESX1* fold-change ratio between *NFKB2* KI and WT at d48 was of 2.06 (p=0.22). However, the type of representation for expression kinetics (values relative to the expression peak in WT) and the scale used made it difficult to see. In the new version of the manuscript, we analysed more samples from the same experiments, and new figure (now 6B) shows significant increase of *HESX1* expression (Fc = 2.46, p=0.019) in *NFKB2* KI.

Also, qPCR results come from at least two different experiments whereas RNAseq come from a single one. For RT-qPCR, 6 HPOs per genotype were picked and further analysed. As we found that only 60-70% of organoids show signs of pituitary cell differentiation, we chose to perform a preselection of organoids, based on RT-qPCR expression of selected markers (*SOX2*, *HESX1*, *PITX1*, *LHX3*, *TBX19*, *POU1F1* and *POMC*) in order to avoid having “empty” HPOs sent for bulk RNAseq. We compared *HESX1* expression ratios obtained by the two different techniques on the same samples (the ones used for RNA-seq) and found values of 2.19 (p=0.03) and 1.83 (p=0.061) for RNA-seq and RT-qPCR respectively. This is illustrated in Supplementary Figure 7. Our new results thus clearly demonstrate the increase in HESX1 expression in NFKB2 KI from d27 to d75.

(4) Throughout the authors focus on POMC gene expression and ACTH antibody immunopositive as being indicative of corticotroph cell identity. In the human fetal pituitary melanotrophs are present and most ACTH antibodies are unable to distinguish these cells from corticotrophs. Is the antibody used specifically for ACTH rather than other products of the POMC gene? It is unlikely that all the ACTH-positive cells are melanotrophs, nevertheless, it is important to know what the proportions of the 2 POMC-positive cell types are. This could be distinguished by looking for the expression of NeuroD1, which would also define whether corticotrophs are committed but not fully differentiated in the NFKB2 mutant organoids. In support of an effect on corticotrophs, it is notable that CRHR1 expression (which would be expected to be restricted to this cell type) is reduced by 84% in bulk RNAseq data (Table 1) and this may be an indicator of the loss of corticotrophs in the model.

The antibody we used is directed against ACTH. In HPOs, PAX7 expression was barely detected during the whole experiment. Moreover, although PCSK2 transcripts were observed, their expression started very early (d27) and remained constant, suggesting that an expression of this gene in hypothalamic cells rather than pituitary cells. All these observations suggest that melanotrophs are very unlikely to be present in HPOs.

(5) Notwithstanding the caveats about whether the organoid model recapitulates in vivo pituitary differentiation (see 1 above) and whether the bulk RNAseq accurately reflects expression levels (see 3 above), there are potentially some extremely interesting changes in gene expression shown in Table 1 which warrant further discussion. For example, there is a 25-fold reduction in POU1F1 expression which may be expected to reflect a loss of somatotrophs in the organoid (and possibly lactotrophs) and highlights the importance of characterising the effect of NFKB2 on other anterior pituitary cell types within the organoid. If somatotrophs are affected, this may be relevant to the organoids as a model of DAVID syndrome as GH deficiency has been described in some individuals with NFKB2 mutations. The huge increase in CGA expression may reflect a switch in cell fate to gonadotrophs, as has been described with a loss of TPIT in the mouse. These are examples of the changes that warrant further characterisation and discussion.

We performed a more in-depth analysis of other pituitary lineages (mainly somatotrophs). We confirmed the strong reduction in *PROP1* and *POU1F1* expression in *NFKB2* KI organoids. Although the strong increase in *CGA* expression in the mutant may raise the possibility of a redirection towards gonadotroph lineage, the lack of change in *NR5A1* expression may suggest otherwise.

These results are now illustrated in figure 12 and discussed in a full paragraph.

(6) How do the authors explain the lack of effect of NFKB2 mutation on global NFKB signalling?

The most likely explanation is that p100/p52 is not involved in controlling the expression of other members of NFKB signalling. Therefore, the absence of global alteration of NFKB signaling pathway shows that mutant p100/p52 protein is directly responsible for the observed phenotype.

**Recommendations for the authors:**

**Reviewing editor summary of recommendation to authors:**
The use of hypothalamic-pituitary organoids can provide a fundamental understanding of pituitary gland development and differentiation. Their use to study human pituitary insufficiency is important, gaining insight into the aetiology of disease and if it implicates the hypothalamus or anterior pituitary. To this end, there is only one other example of their use in the literature, where Matsumoto et al, (2019), used OTX2-mutant hypothalamic-pituitary organoids to understand the aetiology of pituitary hypoplasia driven by OTX2 mutations. This being the second example of using gene editing in human iPSC-derived hypothalamic-pituitary organoids, these studies have improved the efficiency of differentiation previously published by Suga et al. (2011) for ES cells, and Matsumoto et al. (2019) for iPS cells. In addition, it has solidified that this method is useful, especially when studying hypothalamic involvement in human pituitary anomalies, due to the concerted development of these two structures.The reviewers recognise the valuable insight provided into the mechanism of NFKB2 action during pituitary development and how this human organoid model might be one of the few or only approaches suitable to discover the aetiology of the pituitary phenotype.The reviewers agree that both the evidence provided from the organoid model, as well as the characterisation of the phenotype are incomplete. In particular, the strength of evidence would be improved by analysing additional independent clones for both NFKB2 as well as TBX19 gene-edited iPSCs. Additionally, analysis of NFKB2 expression both in vivo and in the organoids, as well as analysis for the NFKB2 targets put forward, would be a lot more informative to help understand this phenotype.The main recommendations discussed are summarised here and the reviewers have elaborated on these points in their individual reviews:The two ACTH-deficient organoid models use a single genetically engineered clone, and the potential for variability amongst clones, unrelated to the mutation, makes the conclusions less compelling. Two independent homozygous clones were obtained for NFKB2 but only one was used, so analysis of the second clone would strengthen the findings. A heterozygous clone was also obtained and given all NFKB2 patients are heterozygous for what appears to be dominant negative mutations, the heterozygous clone ought to be analysed. Analyses of these additional clones would give more strength to the conclusions, showing reproducibility and the effect of mutant gene dosage. The reviewers provide excellent suggestions for alternative controls for the engineered iPSC lines in their specific comments.The effect of TBX19 mutation on early pituitary fate markers LHX3 and PITX1 is surprising given the phenotype of the knockout mice and patients with mutations. If the developmental profile of essential transcription factors does not recapitulate the in vivo expression in this well-characterised mutant, this brings the organoid model into question. Thus, analysis of a further clone for the study of mutant TBX19 would be crucial. The validity of this control affects the interpretations relying on expression differences in the NFKB2-mutant organoids.The study has implicated NFKB2 in pituitary development, but more insight is needed to fully understand disease pathogenesis. The authors presented potential downstream targets of NFKB2 action, including transcription factors and key signalling pathway components; further analyses of NFKB2 expression and experiments investigating the relevant factors in more detail will help elucidate this point.Discerning between the hypothalamus and pituitary tissue is fundamental to interpreting phenotypes: (i) To pinpoint the primary tissue affected by NFKB2 deficiency, staining for NFKB2 during development in vivo will determine if this is expressed both in the developing hypothalamus and anterior pituitary gland or only one of these tissues. (ii) Using markers of hypothalamus and pituitary to discern between these two tissues in organoids, will provide a lot of valuable information where expression changes are presented. This would help discern the contribution of the developing hypothalamus as this is still unclear and has not been discussed. Knowing which tissue compartments NFKB2 is expressed in the organoids would also be of great value.The organoids provide an opportunity to characterise the effects of NFKB2 on other pituitary cell types, since the bulk RNAseq presents intriguing changes indicating that not only corticotrophs may be affected. This may be of relevance to patients, which can have additional pituitary hormone deficiencies. If NFKB2 is expressed in the pituitary, demonstrating expression in the different cell types in vivo as well as in the organoids would help interpret the phenotype. Is this expressed only in corticotrophs/corticotroph precursors, or in additional endocrine cells?

We agree with these considerations and the summary and thank the Editors for their assessment. Although we indeed share the idea that reproduction of the experiments on a second clone would be a useful confirmatory step, we have not been able to reach this goal within a reasonable time frame for the reason mentioned above (unavailability of the main research engineer knowledgeable in the challenging methods involved for organoids differentiation) and due to the long turnaround time of this kind of experiments (3 months for the whole differentiation starting form hiPSC). We therefore decided to publish on a single clone while we are still aiming at reproducing our results on at least a second one and will hopefully be able to provide these additional data in a subsequent revised version. We now acknowledge this limitation in the final part of the Discussion.

We have analysed more samples by RT-qPCR and further analysed RNASeq data from *NFKB2* KI organoids, which provided with more insights into the different levels where *NFKB2* may play a role. Specifically, we now show the effect of *NFKB2* mutation on hypothalamic growth factors and pituitary progenitor differentiation (figure 10), different stages of corticotroph maturation (figure 11) and effects on *PROP1*/*POU1F1*-dependent lineages (figure 12). We confronted our results to publicly available ChIPseq data concerning p52 transcriptional targets (figure 13). We have now provided several additional figures derived from these analyses, including a synthetic figure to summarize the most relevant observed effects (Fig. 14).

**Reviewer #1 (Recommendations For The Authors):**
In organoids, it is essential to stain for NFKB: is it the hypothalamus or the pituitary that expresses NFKB, and if the pituitary, is it the corticotroph itself or the surrounding cells? If immunostaining is not available, FISH or RNAscope can be used to look at expression.

Figure 7 shows stronger expression of p100/p52 in pituitary progenitors, and some expression in the hypothalamic part of the organoid. Due to current lack of biological material and length of experimental procedure, we could not yet determine which differentiated cell types express p100/p52, but this is clearly something we will look at in further experiments.

Regarding Figure 7, NFKB2 (D865G/D865G) shows no LHX3 expression already at day 48. It would be better to look at expression including PITX1 at an earlier time point to see at what point differentiation is impaired.

RT-qPCR results show no statistically significant changes in *PITX1* (Fc=0.58, p=0.25) or *LHX3* (Fc = 0.15; p=0.22) expression at d27, although there was a tendency towards downregulation.

Is it really just a species difference that NFKB2-deficient mice do not have abnormal pituitary function? This needs to be discussed in the manuscript.

_Nfkb2_Lym1/Lym1 mice and *NFKB2* KI model have different but functionally very similar mutations, as they both lead to an abnormal processing of p100 and a strong reduction of p52 content. In mice, these mutations are more severe than the complete absence of Nfkb2 gene product, and they have been called “super repressors”. It is therefore surprising that no pituitary phenotype as been observed in mice. In our opinion, this constitutes a strong argument in favour of an inter-species difference, at least for the pathogenicity of this type of mutations.

This point is now addressed in the Discussion

Just looking at changes in gene expression by qPCR and bulk RNA-seq does not give enough information about localisation. We wish RNA-seq had at least been separated by FACS first. For example, FACS can separate the anterior pituitary and hypothalamus by EpCAM positivity/negativity (PMID: 35903276), so we would like to see gene expression in such separated samples.

This is a pertinent suggestion. We are aware of these techniques and we hope we will be able to include them in future studies

For Figures 2 and 6, just looking at changes in gene expression by qPCR does not provide localisation information, so either (1) immunostaining for LHX3 and NKX2.1 should be shown in each aggregate as in FigS3, or (2) qPCR should be performed on the FACSed cells. (2) qPCR on FACSed cells.

PITX1, LHX3 (as confirmed by our immunofluorescence data) and HESX1 are only expressed in non-neural tissue. TBX19 could be expressed in the hypothalamic part of the organoid, but we observed very little immunostaining outside the outermost layers of organoids (i.e. pituitary tissue). The antibody we used to detect corticotrophs only recognizes ACTH, and therefore only marks pituitary cells.

In addition, pathway and gene ontology analyses should be performed.

Pathways and gene ontology have been performed. However, as organoids consist of two different tissues, the analysis of over 4800 differentially expressed genes did not give us very informative results, apart from an impairment of retinoic acid signalling that we are currently investigating

**Reviewer #2 (Recommendations For The Authors):**
The differentiation of iPSC to organoids could be variable. The authors indicate that 200 organoids were analyzed for each line, and 3-10 organoids were analyzed per time point, genotype, and assay. Is it clear that 100% of the organoids differentiate to produce corticotropes? Please clarify.

In our experiments, almost 90% of organoids give rise to non-neural ectoderm, as demonstrated by PITX1 expression. However, depending on experiments, only 60-70% of organoids give rise to pituitary progenitors (LHX3+) and subsequently to corticotropes. This has been clarified in the text.

For TBX19, it seems surprising that there is an effect on PITX1 and LHX3 expression, since TBX19 expression is normally activated after these genes are expressed. An effect of TBX19 on EMT would also be surprising as the knockout mice do not have dysmorphology of the stem cell niche. The only evidence for an effect is the reduced IHC for E-cadherin. If this is an important point, the authors should examine other EMT markers such as Zeb2. The TBX19 knockout mice appear to form corticotropes based on the expression of NeuroD1, even though they lack TBX19 and POMC expression. It would be reassuring to see that NeuroD1 is normally expressed in the TBX19 mutant organoids.

We agree that the effect of the TBX19 mutant on early pituitary progenitor development is rather puzzling. In our model, TBX19 is expressed throughout the whole experiment, although it is at very low levels in undifferentiated hiPSCs compared to peak expression (over 50-fold difference).

During the CRISPR-Cas9 gene edition, we obtained a clone with a homozygous one base insertion at the cutting site, leading to a frameshift and a premature stop codon 48 bases downstream. This would result in an expected protein of 163 amino acids instead of 488, but with potentially still functional DNA-binding ability. This mutation had a similar effect on *LHX3* and *PITX1* as the *TBX19* KI mutation, although it was even more severe. Our most likely explanation is that the two TBX19 mutants we generated have dominant negative effects. Contrary to mouse, little is known about *TBX19* expression in early human pituitary development, but scRNA-seq data on human embryonic pituitaries (Zhang et al.) show low expression in undifferentiated pituitary progenitors between 7 and 9 weeks of gestation. Therefore, early expression of these dominant negative proteins could perturb differentiation in the organoids. Future development of hiPSCs lines with total absence of TBX19 should help clarify these questions.

Apart from the lack of change in *ZEB2* expression in TBX19 KI (Fc = 1.15; p = 0.35), we did not look further for changes in EMT markers in *TBX19* KI. However, we added a more detailed analysis for EMT markers expression in *NFKB2* KI based on RNAseq results (see table 2).

Due to lack of material, we could not confirm NEUROD1 expression by immunostaining. However, RT-qPCR showed there was no change in *NEUROD1* expression in *TBX19* KI (Fc = 0.81; p = 0.64)

NFKB2 IHC was markedly reduced in NFKB2 D865G/D865G organoids. Based on previous experiments, the mutant protein should be expressed but not activated by proteolytic cleavage. It is possible that the antibody has a different affinity for the mutant protein and/or the uncleaved protein may be unstable. Can this be clarified? The mRNA for mutant NFKB2 appears unchanged in Table 1.

This is puzzling indeed. We did not notice any change in *NFKB2* from d27 to d105, and no significant change either between WT and *NFKB2* KI. Although the antibody we used recognizes both p100 and p52, we cannot rule out the possibility that p100/p52 is degraded by pathways other than proteasome. Another possibility is that p100 interactions with other proteins may decrease the accessibility of the antibody to the epitope

The RNA sequencing data from the NFKB2 organoids is intriguing. It suggests that the NFKB2 mutation may have a modest effect on Tbx19 transcription but not Neurod1. It also suggests there are hypothalamic effects, i.e. altered expression of hypothalamic markers in mutant organoids. Is NFKB2 expressed in the developing hypothalamus? Can normal NEUROD1 IHC be confirmed? It is also intriguing that there may be an effect on EMT. However, there seem to be some discrepancies in the direction of effect on these markers. Please clarify.

This is related to the point just above. P100/p52 is described as a ubiquitously expressed protein. We think that it is expressed in the hypothalamic part of the organoids, but at a lower level compared to pituitary progenitors.

As mentioned before, we could not yet confirm NEUROD1 expression by immunostaining, but RT-qPCR clearly showed there was no change in *NEUROD1* expression in *TBX19* KI (Fc = 0.81; p = 0.64) or NFKB2 *KI* (Fc = 0.88; p = 0.5). However, we investigated other markers of different stages of corticotroph differentiation (see figure 11) and found that the later stages are most affected.

Concerning the EMT, we also found changes in the expression of other markers that are shown in Table 2 and discussed further in the text.

Cytokines have been proposed to play important roles in pituitary differentiation, i.e. IL6. Is there any evidence for an altered cytokine or chemokine expression in the NFKB2 organoids?

We didn’t see any change in IL6 expression NFKB2 *KI* (Fc = 2.34; p = 0.55), but RNAseq shows a strong increase in IL6R (Fc = 8.89; p = 2.13e-09). But at this point, the relevance of these observations remains elusive.

Minor:Some patients with DAVID syndrome have pituitary hypoplasia. The authors measure organoid size and find no differences based on genotype. However, each organoid probably has a variable amount of tissue differentiated to pituitary and hypothalamic fates, therefore, the volume of the whole organoid may not be a good proxy for the amount of pituitary tissue.

We are aware of this issue. However, for most pituitary genes measured by RT-qPCR (*PITX1*, *LHX3*, *TBX19*), the deltaCt values did not drastically vary for a given time point/genotype, suggesting a stable pituitary/hypothalamic ratio.

Figure 9 shows whole transcriptome data for the NFKB2 organoids, and Table 1 lists the data for selected genes. There appears to be disagreement between the significance cut-offs used in the figure and the table. Please adjust.

We removed the fold-change cut-offs to improve clarity

elife120868_0_supp_2945725_rxl2z4. "haft" appears several times, but it should be "half".